# Plasma membrane remodeling in GM2 gangliosidoses drives synaptic dysfunction

Alex S. Nicholson[1], David A. Priestman[2], Robin Antrobus[1], James C. Williamson[3], Reuben Bush[2], Shannon J. McKie[1], Henry G. Barrow[1], Emily Smith[1], Kostantin Dobrenis[4], Nicholas A. Bright[1], Frances M. Platt[2], Janet E. Deane [1]*

1 Cambridge Institute for Medical Research, University of Cambridge, Cambridge, United Kingdom,
2 Department of Pharmacology, University of Oxford, Oxford, United Kingdom, 3 Cambridge Institute for Therapeutic Immunology and Infectious Disease, University of Cambridge, Cambridge, United Kingdom,
4 Dominick P. Purpura Department of Neuroscience, Albert Einstein College of Medicine, Bronx, New York, United States of America

* jed55@cam.ac.uk

## Abstract

Glycosphingolipids (GSL) are important bioactive membrane components. GSLs containing sialic acids, known as gangliosides, are highly abundant in the brain and diseases of ganglioside metabolism cause severe early-onset neurodegeneration. The ganglioside GM2 is processed by β-hexosaminidase A and when non-functional GM2 accumulates causing Tay–Sachs and Sandhoff diseases. We have developed i3Neuron-based disease models demonstrating storage of GM2 and severe endolysosomal dysfunction. Additionally, the plasma membrane (PM) is significantly altered in its lipid and protein composition. These changes are driven in part by lysosomal exocytosis causing inappropriate accumulation of lysosomal proteins on the cell surface. There are also significant changes in synaptic protein abundances with direct functional impact on neuronal activity. Lysosomal proteins are also enriched at the PM in GM1 gangliosidosis supporting that lysosomal exocytosis is a conserved mechanism of PM proteome change in these diseases. This work provides mechanistic insights into neuronal dysfunction in gangliosidoses highlighting that these are severe PM disorders with implications for other lysosomal and neurodegenerative diseases.

## Introduction

Glycosphingolipids (GSL) are enriched in the outer leaflet of the plasma membrane (PM) where they play crucial roles in cell signalling, the immune response and neuronal function [1,2]. GSLs consist of a membrane-embedded ceramide backbone and glycosylated headgroups ranging from simple monosaccharides to larger, branched complex glycans containing different sugar moieties. Complex GSLs containing one or more sialic acids, are known as gangliosides and are highly abundant in the brain

**Data availability statement:** The mass spectrometry proteomics data have been deposited to the ProteomeXchange Consortium via the PRIDE partner repository with the dataset identifier PXD064632 (https://www.ebi.ac.uk/pride/archive/projects/PXD064632). Glycan analysis data, electron microscopy images, CellProfiler protocols and immunofluorescence images have been deposited in the University of Cambridge Data Repository (https://doi.org/10.17863/CAM.118836).

**Funding:** This research was funded by a Wellcome Trust grant (219447/Z/19/Z) awarded to JED that paid the salaries of JED, ASN, HGB and ES (https://wellcome.org). FMP is a Wellcome Trust Investigator (202834/Z/16/Z). The salary of FMP was partially supported by a Royal Society Wolfson merit award (WM130016, https://royalsociety.org). The salary of DAP was partially supported by a Mizutani Foundation for Glycoscience grant (200133, https://mizutanifdn.or.jp). The salary of SJM was supported by a Wellcome Trust Early Career Award (226903/Z/23/Z). The funders had no role in study design, data collection and analysis, decision to publish, or preparation of the manuscript.

**Competing interests:** The authors have declared that no competing interests exist.

**Abbreviations:** GSL, glycosphingolipids; ILVs, intra-lysosomal vesicles; LSDs, lysosomal storage diseases; LRO, lysosome-related organelle; MVB, multi-vesicular body; PMP, plasma membrane proteomics; SCRM, scrambled; SYT1, synaptotagmin 1; WCP, whole cell proteomics.

[3]. GSLs interact with cholesterol and proteins to form dynamic membrane micro-domains that contribute to membrane organization, receptor clustering and vesicle trafficking [4–7]. The specific repertoire and abundance of GSLs in the PM modulates membrane properties and contributes to cell identity [8,9].

The metabolism of GSLs is spatially separated in the cell, with synthesis occurring in the endoplasmic reticulum and Golgi compartments, and degradation occurring in the late endosomal and lysosomal compartments [10,11]. GSL degradation occurs sequentially, with one sugar moiety removed at a time through the action of a series of degradative enzymes. When this process is dysregulated, it causes a range of severe diseases, most of which involve neurodegeneration [3,12,13].

The enzyme β-hexosaminidase A (HexA) is a glycosyl-hydrolase that resides in the lysosome and removes the terminal N-acetyl galactosamine from the ganglioside GM2 (Fig 1A), to yield the ganglioside GM3 [14,15]. The HexA enzyme is a heterodimer of alpha and beta subunits produced by two closely related genes, *HEXA* and *HEXB*, which can only process GM2 ganglioside in vivo when presented by the GM2 activator protein (GM2ap) (Fig 1B) [16,17]. The inability to degrade GM2 causes the severe inherited disorders Tay–Sachs and Sandhoff diseases when *HEXA* or *HEXB* genes are mutated respectively, as the residual homodimers known as HexB and HexS, respectively, are unable to hydrolyze GM2 [18]. Due to the high abundance of gangliosides in neuronal cells, the accumulation of undegraded GM2 within these cells leads to severe neurological dysfunction, neurodegeneration and premature death. Disease severity and age-of-onset is directly correlated with residual enzyme activity, with the most severe forms having effectively no enzyme activity resulting in death by 2 years of age [18,19].

Tay–Sachs and Sandhoff diseases are classified as lysosomal storage diseases (LSDs) due to the presence of enlarged endolysosomes filled with undigested material [20]. In disease cells that show a high accumulation of gangliosides, such as neurons, the endolysosomes are filled with membranous substructures, referred to as whorls or zebra bodies [21]. As endolysosomes are the terminal degradative organelle for both the endocytic and autophagic pathways, defects in their homeostasis have wide reaching effects. The molecular mechanisms that link lysosomal storage to neurodegeneration are incompletely understood but lysosomal dysfunction is a common feature of many neurodegenerative conditions [22].

Several different animal models of Tay–Sachs and Sandhoff disease already exist, including naturally occurring cat, dog and sheep models as well as genetically modified mouse models [23]. Interestingly, mice possess some differences in GSL metabolism compared to humans, as mice have higher levels of a lysosomal neuraminidase that can process GM2 such that both the *HEXA* and *NEU3* genes must be knocked out to produce a Tay–Sachs model better phenotypically resembling human disease [24,25]. The differences in GSL metabolism between mice and humans supports the need to develop human-based models of these diseases for molecular insights into human pathology. iPSC-based neurons and neural progenitor cells have been developed from patient fibroblasts and exhibit lysosomal disturbances and accumulation of GM2, as well as alterations in synaptic exocytosis [26,27]. These

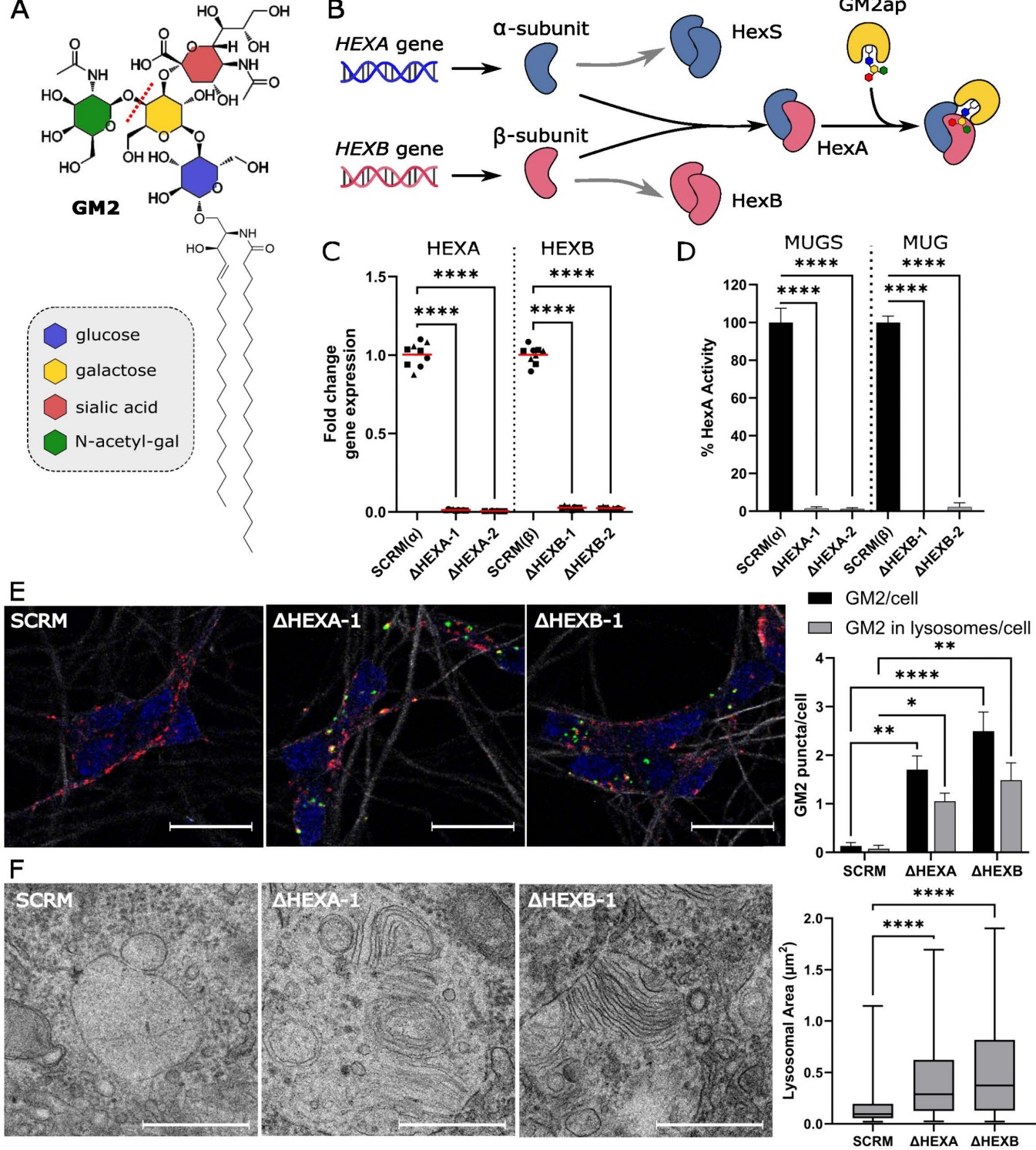

**Fig 1. Neuronal i3N models of Tay–Sachs and Sandhoff diseases.** A. Schematic diagram of the GM2 ganglioside detailing the composition of the glycan headgroup and illustrating which bond is cleaved by the HexA enzyme (red dotted line). B. Schematic diagram of how the α- and β-subunits can form homo- and heterodimers with the HexA heterodimeric isoenzyme being the only one that can cleave GM2. GM2 is presented to HexA by the GM2 activator protein (GM2ap). C. Quantitative PCR (qPCR) analysis of *HEXA* and *HEXB* gene expression in neurons following CRISPRi-induced

knockdown. Fold change relative to SCRM controls are shown for each cell line, $N = 3$ biological replicates (squares, triangles, circles) were carried out in technical triplicate $n = 3$ and the mean is displayed (red line). Statistical significance was determined with a one-way ANOVA, ****$p \leq 0.0001$. D. Activity assays from cell lysates of SCRM and HEXA and HEXB CRISPRi cell lines. Activity was determined using standard fluorescent substrates: MUGS is used for the detection of HexA isoenzyme; and MUG is cleaved by both HexA and HexB isoenzymes. The mean is displayed ± SEM, $N = 3$ biological replicates were carried out and significance was calculated using a one-way ANOVA test, ****$p \leq 0.0001$. E. Fluorescence microscopy images of neurons stained for β3-tubulin (white), LAMP1 for lysosomes (red), GM2 (green) and DAPI (blue). GM2 positive puncta were quantified and analyzed for co-localization with lysosomes. Scale bar (white line) represents 20 μm. Forty-five images were analyzed per cell line across $N = 3$ biological replicates, significance was determined by two-way ANOVA, *$p \leq 0.05$, **$p \leq 0.01$, ****$p \leq 0.0001$. F. Transmission electron microscopy images of SCRM, ΔHEXA and ΔHEXB CRISPRi cell lines. Zoomed images are shown to illustrate membrane whorls and zebra bodies in enlarged lysosomes. Scale bar (white line) represents 500 nm. Twenty images were analyzed per cell line across $n = 5$ EM grids. Only endolysosomes where the entire compartment was visible were quantified. Significance was calculated using a Kruskal–Wallis test, ****$p \leq 0.0001$. Underlying data used to generate these figures are available in S1 Data at https://doi.org/10.17863/CAM.118836.

models have been effectively used to test substrate reduction therapy and to probe disease mechanisms. However, the long differentiation time and genetic variability limit the usefulness of these models for detailed molecular analysis.

Advances in human iPSC technologies allow for new cellular models to probe molecular mechanisms driving disease phenotypes. The i3Neuron (i3N) model is an iPSC line with a stably integrated, doxycycline-inducible NGN2 transcription factor for rapid differentiation into a homogenous, isogenic population of cortical glutamatergic neurons over the course of 14 days [28]. A derivative of this line also possesses a catalytically dead Cas9 (dCas9) enzyme for CRISPR interference (CRISPRi)-mediated knockdown of gene expression [29]. This i3N system allows for the rapid development of neuronal disease models that can be grown on a scale compatible with mass spectrometry based proteomic approaches.

Here we present i3N-based models for Tay–Sachs and Sandhoff diseases, which mimic disease phenotypes including enlarged endolysosomal compartments containing excessive GM2 and membrane whorls. Proteomic analysis identified significant molecular changes, including accumulation of endolysosomal proteins involved in lipid processing as well as several membrane trafficking molecules. These changes result in abnormal accumulation of lipids and proteins at the cell surface impacting neuronal functions and leading to dysregulation of electrical activity. Alterations to the cell surface proteome were also observed in a model of GM1 gangliosidosis. These neuronal cellular models of GM1 and GM2 gangliosidoses represent valuable systems for understanding the molecular mechanisms driving disease pathogenesis.

## Results

### Accumulation of GM2 in endolysosomes of i3N models of Tay–Sachs and Sandhoff diseases

Severe, early-onset forms of Tay–Sachs and Sandhoff diseases possess less than 5% enzyme activity due to large gene deletions or missense mutations [18,19,30]. To develop relevant cellular disease models, efficient CRISPRi-mediated knock down of HEXA or HEXB gene expression, to remove functional HexA enzyme, was required. CRISPRi guides targeting the HEXA or HEXB genes were introduced into the dCas9 derivative of the i3N stem cells (S1 Table). Five cell lines using different guide sequences were made from this parent dCas9 cell line: ΔHEXA-1 and ΔHEXA-2 to model Tay–Sachs disease, ΔHEXB-1 and ΔHEXB-2 to model Sandhoff and a scrambled (SCRM) non-targeting guide to be used as a control. Quantitative PCR (qPCR) analysis of neurons at the reported mature time point of 14 days post doxycycline-induced expression (dpi) of the NGN2 gene confirmed highly efficient knockdown of HEXA and HEXB gene expression. mRNA levels in all cell lines were between 0.5% and 3% of gene expression compared with SCRM lines (Fig 1C). Enzyme activity assays are used to screen carriers and identify disease severity, and here they were adapted to determine activity in lysates of differentiated neurons [31,32]. This confirmed that the loss of gene expression translated into an equivalent lack of functional α and β subunits, as shown by the loss of HexA activity in the respective cell lines (Fig 1D). The levels of enzyme activity shown are consistent with those seen in patients with infantile or juvenile forms of early-onset Tay–Sachs and Sandhoff diseases [18].

The most striking phenotype in cells with HexA dysfunction is lysosomal enlargement caused by the storage of undigested GM2 lipid substrate. Immunofluorescence microscopy using an anti-GM2 antibody revealed accumulation of GM2 in ΔHEXA-1 and ΔHEXB-1 cell lines. The majority of this ganglioside colocalized with LAMP1 positive structures, whilst very little GM2 was detected in the SCRM line (Fig 1E). Analysis of the ultrastructure of ΔHEXA-1 and ΔHEXB-1 cells using transmission electron microscopy revealed significant enlargement of endolysosomal compartments and accumulation of membranous substructures in the form of whorls and zebra bodies, absent in the SCRM line (Fig 1F). This confirmation that GM2 accumulates in endolysosomes and that these compartments contain large amounts of multi-lamellar membranous structures similar to those observed in human and animal disease cells [20,21,23,33,34] supports the validity of these i3N models of Tay–Sachs and Sandhoff diseases.

Previously published work has confirmed that i3N cells express markers of functionally mature neurons by 14 dpi [28]. qPCR analysis of the i3N cell lines confirmed expression of neuronal markers (synaptophysin SYP, MAP2 and β3-tubulin) and loss of stem cell markers (NANOG and OCT-4) at this time point (Figs 2A and S1). These data confirm that the disease model lines can still differentiate normally and express relevant markers of neuronal maturity.

The quantification of complex glycosylated sphingolipids in cells is challenging using mass spectrometry approaches due to several glycan moieties possessing the same molecular mass [35]. An alternative strategy to accurately quantify the abundance of the different ganglioside headgroups is known as glycan profiling [36,37]. This involves extraction of all GSLs from cell lysates, cleavage of the ceramide tails using endocerebrosidase (ceramide glycanase), fluorescent labelling of the glycan headgroup with 2-Anthranilic Acid and separation of the labelled glycans using HPLC for quantification of individual peaks against known standards (S2 Fig). To determine how much GM2 was accumulating and whether any other gangliosides were altered in these disease models, glycan profiling was employed to quantify the abundance of all ganglioside species (Fig 2B). These data confirm substantial accumulation (>20-fold) of GM2 in ΔHEXA and ΔHEXB neurons, compared to SCRM and demonstrates that no other gangliosides are significantly altered in their abundance by the accumulation of GM2.

These data from i3N disease models match closely to the observed phenotypes in diseased patient neurons [21]. Specifically, the accumulation of GM2 lipid substrate, particularly in the late endolysosomes, causing formation of membrane whorls and enlargement of this compartment. Therefore, this is a valid model for probing the molecular mechanisms driving disease pathology.

## Impact on the proteome of GM2 accumulation in neurons

To understand the impact that GM2 accumulation has on neuronal function, whole cell proteomics (WCP) was carried out on SCRM, ΔHEXA-1/2 and ΔHEXB-1/2 cell lines at 14 dpi. High confidence targets were selected based on a significant change in abundance of at least 1.2-fold compared to SCRM control. When compared with each other, the models of Tay–Sachs (ΔHEXA-1/2) and Sandhoff (ΔHEXB-1/2) disease cell lines showed very few significant differences in protein abundances, indicating that the changes observed, compared to the SCRM line, are due to GM2 accumulation specifically rather than the exact gene/subunit that is targeted (S3 Fig). Thus, for analysis, data from both the models of Tay–Sachs (ΔHEXA-1/2) and Sandhoff (ΔHEXB-1/2) were combined and compared to SCRM controls. All GM2 gangliosidosis disease models showed significant accumulation of endolysosomal proteins including degradative enzymes (such as LIPA and CTSD), membrane pumps (SLC37A3 and LMBRD1), lipid transporters (NPC1/2, SCARB2 and ABCA3), integral limiting membrane proteins (LAMP1), modulators of lysosomal position (RNF167 and TMEM106B) and exosomal markers (TSPAN3/6/7 and CD63) (Fig 2C and S2 Table). qPCR data for five proteins involved in the TFEB-mediated CLEAR response (Coordinated Lysosomal Expression And Regulation) that drives lysosomal biogenesis showed no significant changes between control and knockdown lines. Thus, these changes in lysosomal protein abundance were primarily due to the accumulation or reduced turnover of endolysosomes rather than increased lysosomal biogenesis (S4 Fig) [38,39].

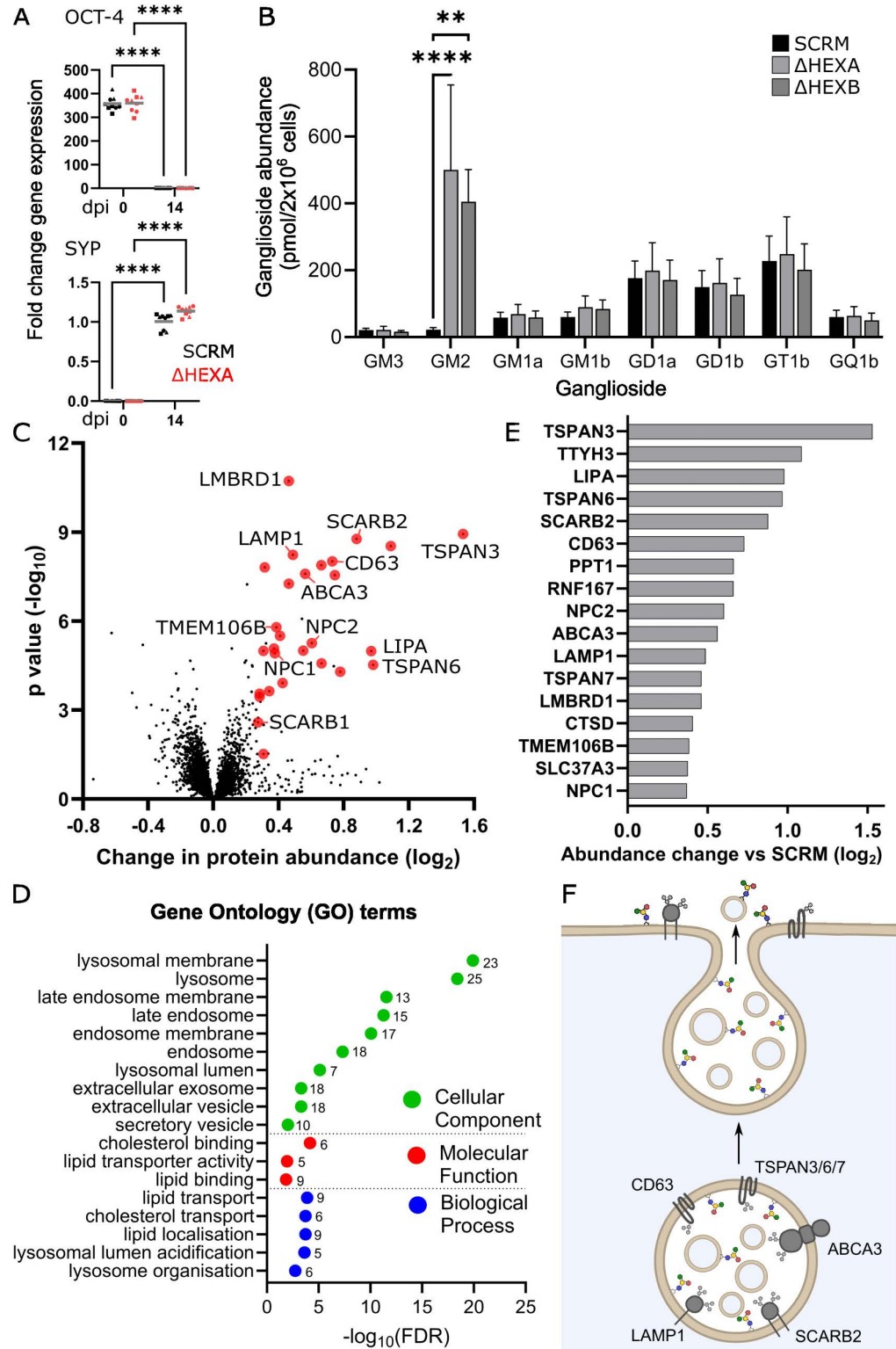

**Fig 2. Neuronal maturation, GM2 quantification and the impact of GM2 accumulation on the whole cell proteome of Tay–Sachs and Sandhoff disease models. A.** qPCR analysis of gene expression for the stem cell marker OCT-4 and the neuronal marker synaptophysin (SYP) in SCRM and ΔHEXA-1 cell lines at 0 and 14 dpi. Fold change is calculated relative to 14 dpi SCRM controls, $N = 3$ biological replicates were carried out in technical

triplicate $n = 3$ (squares, triangles, circles) and the mean is displayed (grey line). Significance was determined with a two-way ANOVA, ****$p \leq 0.0001$. B. Quantification of whole-cell gangliosides at 14 dpi for SCRM, ΔHEXA and ΔHEXB cell lines, the mean of $N = 3$ biological replicates is displayed ± SEM. Significance was determined by two-way ANOVA, **$p \leq 0.01$, ****$p \leq 0.0001$. C. Quantitative whole cell proteomics (WCP) data for ΔHEXA-1/2 and ΔHEXB-1/2 neurons compared with the SCRM control. A volcano plot is shown with average fold change ($x$-axis) across $N = 3$ biological replicates and significance ($y$-axis, two-sided $t$ test) across the three replicates. Endolysosomal proteins are colored in red with targets mentioned in the text labelled. D. Gene ontology (GO) term analysis for proteins significantly changed in the WCP dataset. Changes are shown for cellular component, molecular function and biological process with the change plotted as the false discovery rate ($\log_{10}$FDR) and the number of proteins in each group indicated. E. Select targets from the WCP are represented graphically to illustrate the fold change in whole cell protein abundance in ΔHEXA and ΔHEXB neurons vs. SCRM neurons. F. Schematic diagram of lysosomal exocytosis and how this process can contribute to changes in lipid and protein abundance at the plasma membrane (PM). A selection of lysosomal proteins increased in abundance in the WCP of ΔHEXA and ΔHEXB neurons are illustrated. Underlying data used to generate these figures are available in S1 Data at https://doi.org/10.17863/CAM.118836.

The proteomic changes seen in these cells have been driven by lipid accumulation and the subsequent downstream effects that this accumulation has upon lysosomal function and cellular trafficking. Several of the proteins identified here, such as the tetraspanins (TSPANs), have recently been shown to directly bind GSL headgroups [40] and several others such as LIPA, PPT1, ABCA3 and NPC1/2 are directly involved in lipid and sphingolipid processing [41–44]. Functional annotation and enrichment analysis using the Database for Annotation, Visualization and Integrated Discovery (DAVID) identified several gene ontology (GO) terms that were highly enriched in this dataset including lysosomal, endosomal and membrane cellular component terms (Fig 2D). This analysis also highlighted significant enrichment of proteins specifically involved in lipid and cholesterol binding and transport. This suggests that some of the proteins that are accumulating in these diseases are specifically products of lipid accumulation rather than a product of general lysosomal dysfunction. In further support of this, 390 lysosomal proteins including V-type ATPases (ATP6 family), mannose-6-phosphate receptor (M6PR) and biogenesis of lysosomal organelle complex subunits (BLOC1) are quantified in the WCP but are not significantly increased in abundance. This observation of altered organelle composition, rather than general accumulation, is consistent with recent studies demonstrating that endolysosomal compartments are not all the same, but can be dynamically remodeled and possess substantial heterogeneity in their protein composition [45,46].

Of particular interest in the WCP datasets was the increasing abundance of a subset of proteins involved in endolysosomal trafficking and exocytosis as well as exosome formation and release including the TSPANs 3, 6, and 7, CD63 and ABCA3 (Fig 2E and 2F) [47–49]. The exocytosis of endolysosomal components has a normal role in membrane repair and secretion but is also a strategy used by the cell to attempt to clear accumulating cell debris [50]. The fusion of these compartments with the PM may be beneficial for the cell and has been proposed as a mechanism that could be enhanced and exploited to treat LSDs [51]. However, the delivery of endolysosomal proteins and lipids to the PM may also result in undesired changes at the cell surface that may interfere with cellular function.

## Consequences of GM2 accumulation on the plasma membrane proteome

The increased abundance of cellular proteins involved in lysosomal trafficking and exocytosis may have significant consequences for the protein composition of the PM. The proteome of the PM makes up a small fraction of the total cellular proteome meaning that changes at the PM are often not quantifiable using whole-cell analytical techniques [52]. To ensure accurate quantification of the PM proteome, PM proteins were specifically labelled with aminooxy-biotin before cell lysis allowing for enrichment with streptavidin prior to mass spectrometry analysis. As before, all four cell lines, ΔHEXA 1/2 and ΔHEXB 1/2 were combined and compared against the control SCRM cell line at 14 dpi with high-confidence targets selected based on a significant change in abundance of at least 1.2-fold compared to SCRM control. Functional annotation of GO Terms was used to define cellular compartments and the data consisted overwhelmingly of membrane proteins, though some membrane adjacent proteins were also detected.

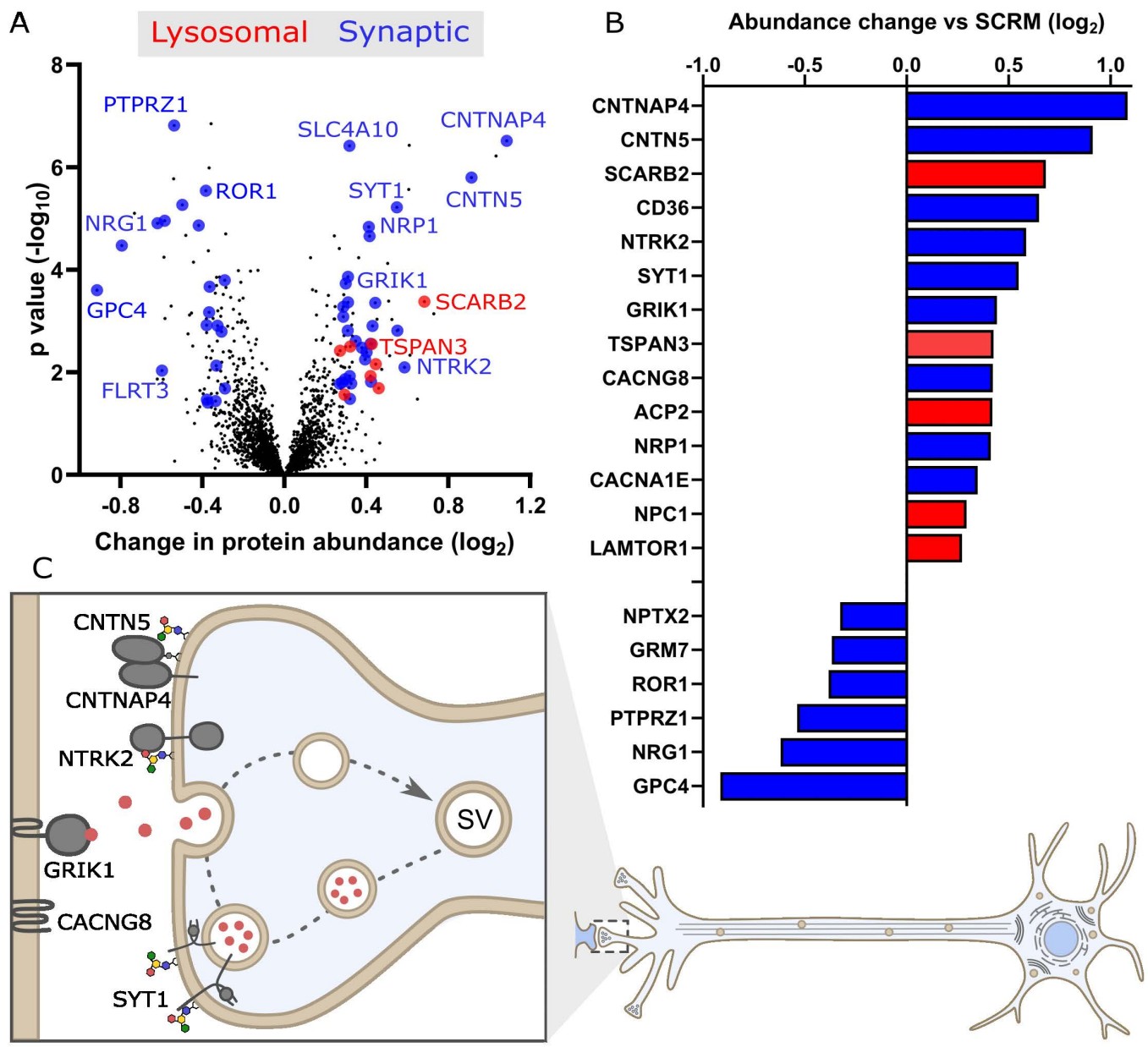

**Fig 3. Molecular consequences of GM2 accumulation on the protein composition of the plasma membrane (PM).** A. Quantitative mass spectrometry following enrichment of PM proteins from ΔHEXA and ΔHEXB neurons compared with the SCRM control. A volcano plot is shown with average fold change (x-axis) across three biological replicates and significance (y-axis, two-sided t test) across the three replicates. Targets colored in blue are synaptic proteins and in red are lysosomal proteins with selected targets labelled. B. Select targets are represented graphically to illustrate the fold change in PM protein abundance in ΔHEXA and ΔHEXB neurons vs. SCRM neurons. C. Illustration of proteins with increased abundance at the PM in ΔHEXA and ΔHEXB neurons, compared to SCRM neurons, that play important roles in neuronal signalling such as synaptic vesicle recycling and synaptic adhesion and receptor molecules. Underlying data used to generate these figures are available in S1 Data at https://doi.org/10.17863/CAM.118836.

Significant increases in the abundance of several endolysosomal proteins were observed at the PM including proteins present in the limiting membrane including SCARB2, ACP2, NPC1 and TSPAN3 (Fig 3A, 3B and S3 Table). This provides strong evidence that endolysosomal compartments are fusing with the PM to disgorge undegradable lipid material, further supported by the increase in abundance of the lipid transporter CD36.

Importantly, functional classification of the PM proteome data also revealed substantial additional changes at the PM including altered abundance of several synaptic proteins including "lipid raft"-associated and GPI-anchored proteins, such as CNTN5, CNTNAP4 (also known as Caspr4), IgSF21 and several glypicans GPC4/5/6 (Fig 3B and 3C). The protein abundance changes observed here at the PM are likely driven by mislocalization, mistrafficking or altered degradation of proteins rather than altered gene expression as demonstrated by qPCR data for several high confidence targets (S5 Fig). Accumulating GM2 at the PM may be altering the formation or stability of membrane microdomains [53]. These membrane domains are critical for synaptic function including the clustering of synaptic proteins and typically contain large amounts of complex gangliosides that often interact directly with synaptic proteins [3,54,55]. Even small changes to the relative abundances of GM2 in these membrane microdomains may have a large impact on synaptic signalling and thus neuronal function.

In addition to proteins known to associate with membrane microdomains generally, some of the proteins identified here are direct ganglioside-binding proteins. The receptor tyrosine kinases NTRK2/3 (also known as TRKB/C) are significantly increased in abundance at the PM. NTRK2/3 promote neurite outgrowth, axonogenesis and synaptogenesis and also directly associate with gangliosides at the PM, typically GM1 [56]. NTRK2/3 are receptors for brain-derived neurotrophic factor (BDNF) and neurotrophin-3 [57] respectively and the increased abundance of these proteins at the PM in these disease models may be driving synaptogenesis and modulating neuronal activity.

Calcium flux across the PM and organelle membranes is known to be disrupted in both GM2 and GM1 gangliosidoses. Calcium sequestration is defective in both the ER and the lysosome in these disorders [58–61]. Several membrane proteins important in calcium signalling pathways are increased in abundance at the PM in these models of GM2 gangliosidoses including CACNA1E, CACNG8 and synaptotagmin 1 (SYT1). CACNA1E is a subunit of the 2.3 voltage gated calcium channel, driving calcium influx into cells at the pre-synapse to facilitate synaptic vesicle (SV) release, whilst CACNG8 positively regulates postsynaptic AMPAR activity [62–64]. The protein SYT1 is a calcium sensor required for neurotransmitter release and SV recycling at the synapse [65,66]. SYT1 and the SYT1-binding protein synaptic vesicle glycoprotein 2b (SV2b) [67,68] are both significantly increased in abundance at the PM in HEXA deficient cells. Previously, increased abundance of SYT1 at synapses was shown to drive faster SV internalization and increased SV exocytosis [69]. Similarly, the increase in abundance of SYT1 in these models of Tay–Sachs and Sandhoff diseases is likely to significantly impact synaptic signalling. Interestingly, SYT1 also specifically binds to the ganglioside GT1b suggesting there may be a direct link between ganglioside dysregulation and SV cycling [70].

Additional evidence of changes at the PM that are driven by altered GM2 metabolism and that may influence synaptic signalling in these neurons is the increased abundance of KCNA2, SCN2b and GRIK1 which are voltage and neurotransmitter gated potassium and sodium channels [71–73]. These and other observed abundance changes in synaptic proteins in the PM proteomics data cannot distinguish whether these proteins are enriched within synapses or more generally distributed across the PM. As ganglioside-containing membrane microdomains are enriched at synapses [3,54,55] it is reasonable to hypothesize that some of these changes will be localized to the synapse. Similarly, the lysosomal proteins at the PM may be clustered at specific locations or may be spread across the PM. Whether localized to the synapse or not, several of the PM proteins that are altered in abundance are involved in synaptic signalling suggesting that these changes may have a direct impact on the electrical activity of the disease neurons.

## Molecular changes at the PM drive hyperactivity of Tay–Sachs neurons

To probe the impact of the PM proteome changes on neuronal signalling, initial measurements were made using a fluorescent calcium sensor protein [74]. Interestingly, this revealed that whilst SCRM, ΔHEXA-1 and ΔHEXB-1 neurons are beginning to fire spontaneously by 14 dpi, they do not undergo spontaneous synchronized signalling until 21−28 dpi (Fig 4A). This suggested that later time points might be more relevant for the study of mature synapse formation. Thus, although the neurons express markers of maturity by 14 dpi, it is not until 28 dpi that the neurons are functioning as a

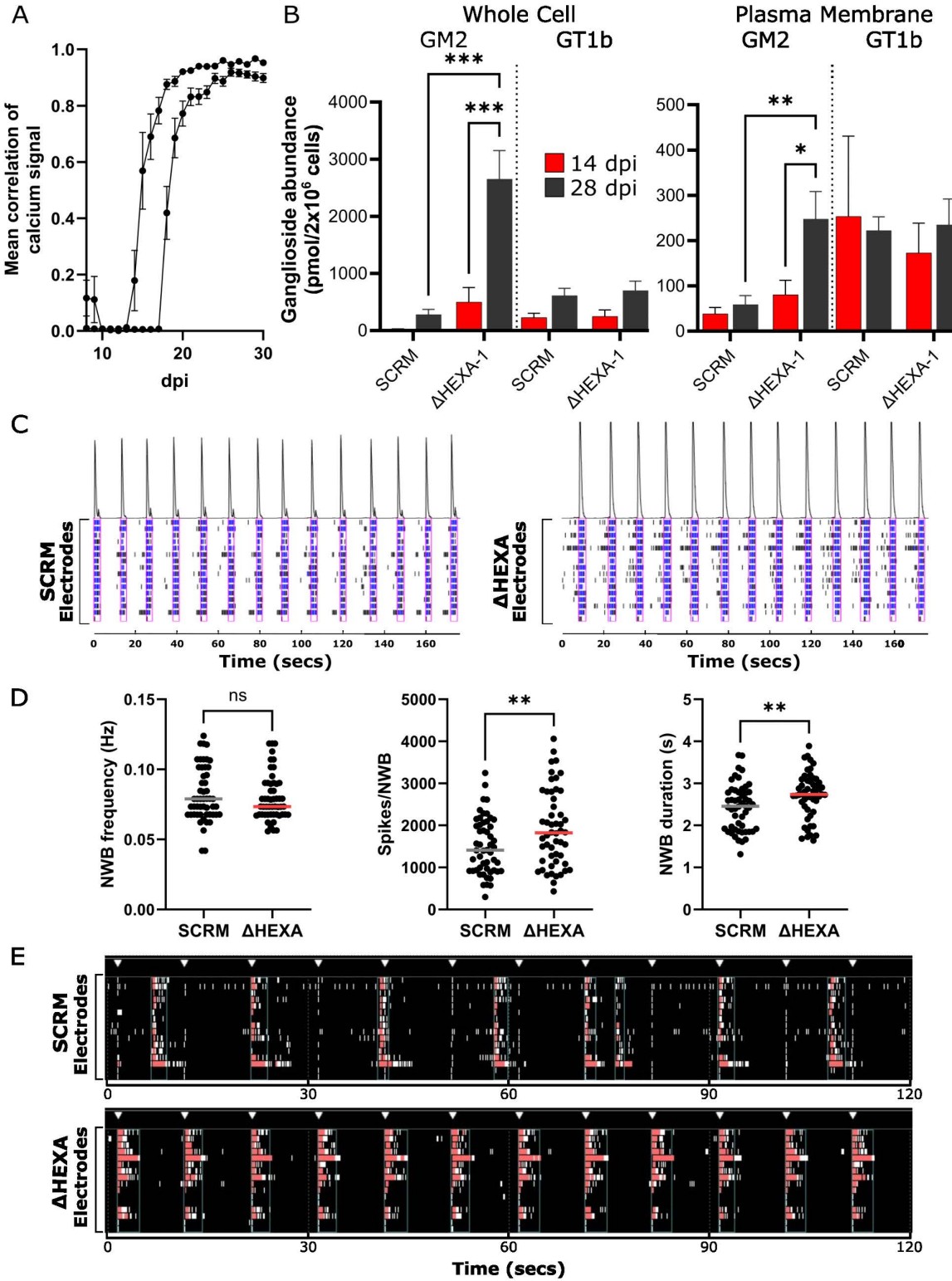

**Fig 4. Increased spontaneous and stimulated neuronal network activity in ΔHEXA neurons compared with SCRM neurons.** A. Calcium signalling was monitored over 30 dpi for SCRM, ΔHEXA and ΔHEXB cell lines. Calcium signalling of cells starts to become synchronous (measured as correlation of bursting objects) from about 15 dpi with increasing signal correlation over time. Four technical repeats, $n=4$ of $N=2$ independent experiments are

shown with data from all cell lines (mean of SCRM, ΔHEXA and ΔHEXB ± SEM) combined to demonstrate that within an experiment, all cell lines are well correlated but between experiments, time to 100% correlation can differ until about 25 dpi. B. Quantification of gangliosides GM2 and GT1b from whole cell (left) and PM-enriched (right) samples of neurons at 14 and 28 dpi for $N = 3$ biological replicates of SCRM and ΔHEXA cells. Significance was determined by two-way ANOVA, *$p \leq 0.05$, **$p \leq 0.01$, ***$p \leq 0.001$. C. Representative activity traces (raster plots) of spontaneous neural activity for SCRM (left) and ΔHEXA (right) neurons. Activity is recorded over time on each of the 15 electrodes with spikes (black), bursts (blue) and network bursts (pink boxes) shown over the same 180 s time period. The intensity of network activity is represented as a peak (above). D. Analysis of neuronal activity including network burst (NWB) frequency, number of spikes per NWB and NWB duration for data collected from $N = 3$ biological replicates across $n = 52$ wells over a time window from 38 to 45 dpi. Mean is displayed for SCRM (grey line) and HEXA (red line) and significance was determined with an unpaired $t$ test, **$p \leq 0.01$. E. Representative activity traces (raster plots) for SCRM (upper) and ΔHEXA (lower) neurons in response to repetitive field stimulation (white triangles). Activity over time is recorded on each of the 15 electrodes. Network bursts (red) are shown over the same 120 s time period. Underlying data used to generate these figures are available in S1 Data at https://doi.org/10.17863/CAM.118836.

network. Over this longer time period, it is likely that GM2 accumulates to even greater quantities and the lysosomal exocytosis observed by 14 dpi suggests that this accumulation may also manifest at the PM. To probe alterations of the lipid composition of the PM, the ganglioside-profiling technique used previously (Fig 2B) was modified to include an initial labelling step prior to cell lysis to distinguish cell-surface GSLs from intracellular GSLs. Using a modification of the technique used for labelling PM proteins, the sialic acid moieties of GSLs were labelled with an aminooxy-biotin that altered the elution profile of the cleaved headgroups during HPLC-based separation (S6A Fig). Initial attempts to enrich for the PM fraction of lipids using streptavidin affinity were not successful meaning that the full repertoire of GSL content at the PM could not be determined. However, the quantification of several GSLs, including GM2, was possible as the aminoxy-biotin-labelled GSLs eluted in a region of the chromatogram free from other GSL peaks (S6B Fig). Interestingly, this analysis demonstrated significant accumulation of GM2 at the cell surface of ΔHEXA cell lines, while other quantified gangliosides remained unchanged (Fig 4B). By 14 dpi, the abundance of GM2 at the PM in ΔHEXA-1 cells is twice that in the SCRM cells (81 versus 38 pmol/$2 \times 10^6$ cells) and by 28 dpi increases to 5-fold higher than SCRM cells (248 versus 59 pmol/$2 \times 10^6$ cells). GM2 is not normally present in high amounts at the PM of mature neurons, and this increase in ΔHEXA-1 cells by 28 dpi brings its abundance up to the level of more common neuronal gangliosides such as GT1b.

To explore the impact of changes in GM2 abundance and altered synaptic protein composition of the PM on neuronal activity, multi-electrode array (MEA) experiments were carried out on ΔHEXA-1 and SCRM neurons. MEA allows for the monitoring of both spontaneous signalling of neurons and response to electrical stimulation. Following optimization of the coating strategy and cell seeding, neurons were cultured on 24-well MEA plates to record neuronal activity. Electrical signalling was monitored via an array of electrodes with each electrode measuring spikes of field potential, and when >5 spikes were measured on a single electrode within 100 ms these were considered bursts. When >70% of electrodes in a well registered bursts within 100 ms this was considered a network burst. Consistent with the calcium signalling measurements, neurons were electrically active (bursting) by 14 dpi and firing synchronously (network bursting) by 21 dpi, with this burst strength increasing over time. Data collected between 38 and 45 dpi show that ΔHEXA-1 neurons had significantly more spikes per network burst and had increased network burst duration compared to the SCRM control cells (Fig 4C and 4D). Representative activity traces (raster plots) demonstrate this altered electrical activity in ΔHEXA-1 cells and quantification over three independent experiments reveals significant differences in the network burst behavior.

The longer network bursts seen in the spontaneous electrical activity of ΔHEXA-1 neurons are indicative of more excitation as it takes longer to shut down the burst, suggesting they may have a reduced threshold for electrical firing [75,76]. To test whether this reduced threshold was recapitulated with field stimulation, neurons were subjected to electrical stimulation of 150 mV for 200 μs at a rate of 0.1 Hz for 5 min. The ΔHEXA-1 neurons again responded differently to field stimulation when compared with SCRM neurons. Control neurons did not respond readily to stimulation and continued to fire spontaneously at the same rate as they had prior to the applied stimulation (Fig 4E, white triangles denote applied stimulation). In contrast, the ΔHEXA-1 cells consistently depolarize in synchronicity with the stimulation throughout the entire time course (Fig 4E). The electrical stimulation was increased to 250 mV for 400 μs with the cell lines maintaining

the same behavior, that is ΔHEXA-1 neurons depolarizing with each stimulation and SCRM neurons continuing to sponta-neously fire as they had without stimulation. These data support that the ΔHEXA-1 neurons have a reduced threshold for depolarization compared to SCRM cells. This hyperactive electrical signalling for the ΔHEXA-1 neurons is consistent with the altered abundance at the PM of key synaptic signalling proteins and it is tempting to speculate that these changes are functionally linked.

**Lysosomal exocytosis is a conserved mechanism of PM proteome change in GM1 gangliosidosis**

To determine whether lysosomal exocytosis is a more general consequence of lysosomal accumulation of undigested GSLs, an additional sphingolipidosis model was generated. Defects in GLB1 cause GM1 gangliosidosis where the lipid GM1 accumulates in cells. An iPSC neuron model of this disease was generated by CRISPRi-mediated KD of GLB1. The loss of *GLB1* gene expression was confirmed via q-PCR and GSL quantification revealed significant accumulation of GM1 in neurons (Fig 5A and 5B). Based on the observation that electrical signalling was synchronized by 28 dpi (Fig 4A), SCRM, ΔHEXA and ΔGLB1 cells were grown to 28 dpi for proteomic analysis. WCP for both the GM1 and GM2 disease models reveal several shared proteomic changes in these models (Fig 5C and S4 Table). Analysis of both gangliosido-ses versus SCRM control cells confirmed significant enrichment of lysosomal proteins, particularly those involved in lipid and cholesterol processing (Fig 5D, 5E, and S4 Table). Several of the shared changes were also evident at 14 dpi in the ΔHEXA lines (Fig 2E) and are further increased by 28 dpi including the lysosomal limiting membrane proteins TTYH3, TSPAN3/7, NPC2, ABCA3 and LAMP1. Several lysosomal luminal proteins are also significantly changed in both models by 28 dpi including PSAP, PPT1, CTSD and GNS. Interestingly, additional proteins involved in endolysosomal traffick-ing are also significantly increased by 28 dpi including GGA2, KCTD7, LAMTOR3 and ARL8a supporting that there is a shared and evolving lysosomal phenotype in these models [77–81].

PMP was carried out at 28 dpi for both GM2 and GM1 gangliosidosis models. Analysis of the changes in both models versus control cells reveals highly significant increases in the abundance of several lysosomal proteins at the PM includ-ing TSPAN3, SCARB2 and TTYH3 (Fig 5F and S5 Table). This supports that lysosomal exocytosis may be a general mechanism for disposal of accumulating material from human cells with lysosomal dysfunction. Furthermore, it supports that some of the lysosomal limiting membrane proteins are not efficiently recycled back into endolysosomes resulting in their accumulation at the PM. Within the list of proteins significantly increased at the PM were additional proteins involved in lysosomal trafficking, sorting and exocytosis including PLD3, Rab7a and DNAJC5 [82–85].

## Discussion

The data presented here demonstrate the utility of the i3N cells for modelling neuronal diseases. The ΔHEXA and ΔHEXB neurons faithfully recapitulate the key cellular disease phenotypes and are robust and reproducible enough to be used at scale to produce proteomic datasets to probe the mechanisms of GM2 gangliosidosis disease pathogenesis. The data establishes that whilst the lysosome is clearly the primary affected organelle in the GM2 gangliosidoses, the impact of lysosomal accumulation of lipids spreads beyond this organelle, with important implications for cellular and synaptic signalling in neurons. The changes observed at the PM are in part triggered by lysosomal dysfunction and impact both the lipid and protein composition of this membrane in complex ways. One of the major consequences of these molecular changes is altered synaptic signalling in disease neurons with them exhibiting hyperactivity due to a lower threshold for depolarization.

The immunofluorescence and electron microscopy imaging for the HEXA-deficient cells revealed significant accu-mulation and enlargement of endolysosomal compartments. Interestingly, the more detailed analysis of these changes via WCPs revealed specific enrichment of lysosomal proteins involved in lipid processing and transport, including sev-eral proteins that bind directly to cholesterol and/or glycosphingolipids. This specific accumulation of proteins such as SCARB2, LIPA and NPC1/2 may represent "secondary" changes in response to GM2 accumulation. This suggests that

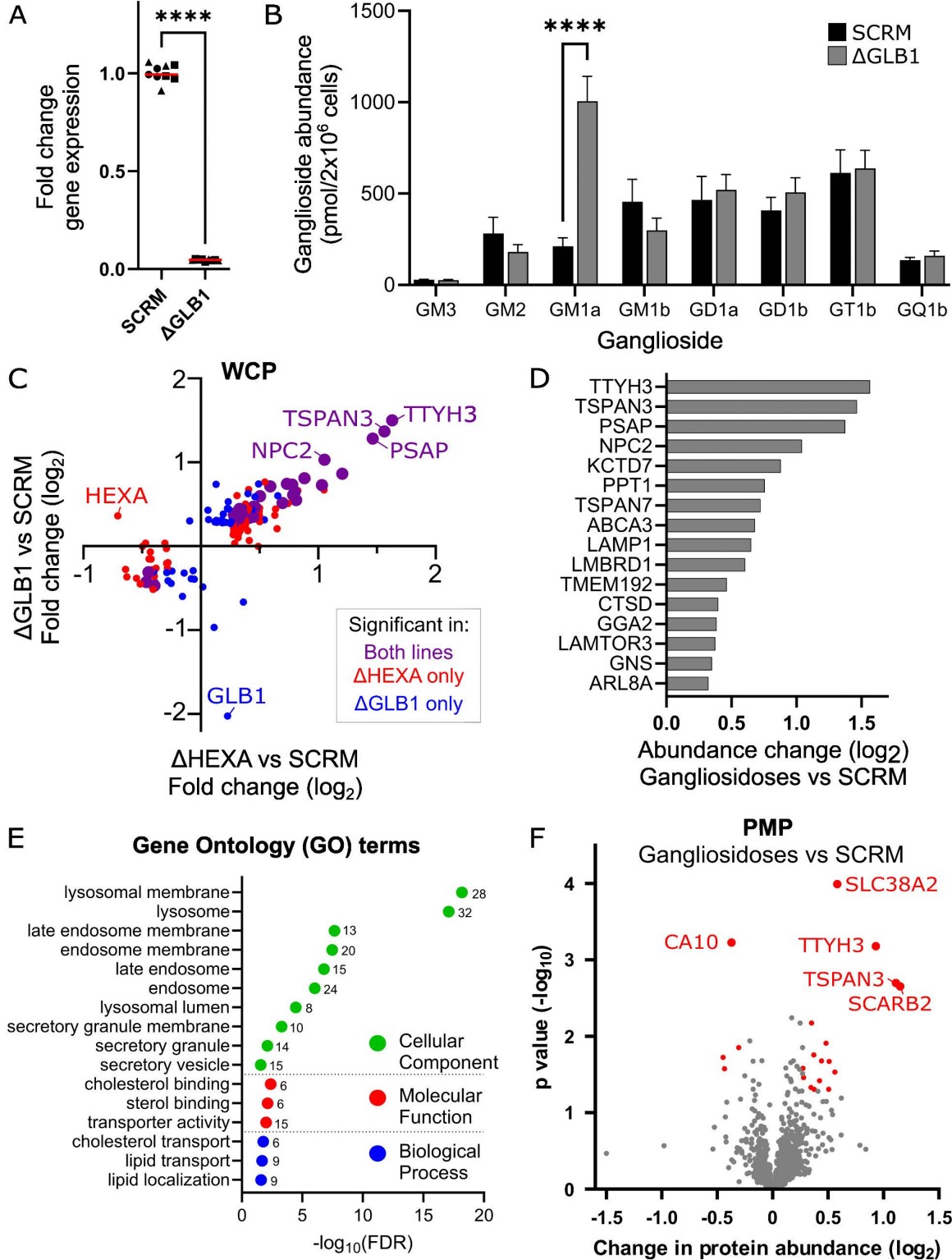

Fig 5. A model of GM1 Gangliosidosis shares lysosomal proteomic changes and hallmarks of lysosomal exocytosis. A. Quantitative PCR (qPCR) analysis of *GLB1* gene expression in neurons following CRISPRi-induced knockdown at 14 dpi. Fold change relative to SCRM controls are shown for N = 3 biological replicates carried out in technical triplicate n = 3 (squares, triangles, circles) and the mean is displayed (red line). Significance

was determined with an unpaired *t* test, ****$p \leq 0.0001$. B. Quantification of whole-cell gangliosides at 28 dpi for SCRM, ΔGLB1 cells, the mean of $N = 3$ biological replicates is displayed ± SEM. Significance was determined by two-way ANOVA, ****$p \leq 0.0001$. C. Analysis of quantitative WCP data at 28 dpi for ΔHEXA neurons compared with SCRM cells vs. ΔGLB1 neurons compared with SCRM cells. Fold change in protein abundance for ΔHEXA cells (*x*-axis) are plotted against fold change in protein abundance for ΔGLB1 cells (*y*-axis). Significance (2-sided *t* test across $N = 3$ biological replicates) is illustrated as those significantly changed in ΔGLB1 only (blue), ΔHEXA only (red) or in both (purple, large data points). D. Select shared targets from the WCP when both gangliosidosis models are combined are represented graphically to illustrate the fold change in whole cell protein abundance. E. GO term analysis for proteins significantly changed in the WCP for both gangliosidosis models is shown for cellular component, molecular function and biological process with the change plotted as the false discovery rate ($\log_{10}$FDR) and the number of proteins in each group indicated. F. Quantitative mass spectrometry following enrichment of PM proteins from ΔHEXA and ΔGLB1 gangliosidosis models compared with the SCRM control. A volcano plot is shown with average fold change (*x*-axis) and significance (*y*-axis, two-sided *t* test) across $N = 3$ biological replicates for each of the ΔHEXA and ΔGLB1 lines compared to $N = 2$ SCRM. Proteins that are significantly changed ($p \leq 0.05$) are colored (red). Underlying data used to generate these figures are available in S1 Data at https://doi.org/10.17863/CAM.118836.

rather than a general blockade of all lysosomal function leading to a wholesale increase in lysosomal components, there are specific changes affecting proteins involved in lipid and cholesterol binding and transport. This observation is consistent with recent studies demonstrating that endolysosomal protein composition can be altered in response to disease or drug treatment and that these compartments demonstrate significant heterogeneity [45,46,86]. The lysosomal lipid processing proteins may be accumulating due to reduced turnover in an attempt to restore lipid homeostasis, or they may be partitioned within the membrane whorls in the lysosomes reducing their exposure to normal degradative processes. The continued accumulation of GM2 and consequential impact on endolysosomal function also appears to trigger lysosomal exocytosis and exosome release as evidenced by increased abundance of trafficking and exosomal proteins in the WCP data and appearance of lysosomal proteins in the PMP datasets. These processes are mechanisms by which the cell attempts to relieve the lysosomal burden of accumulating substrate into the extracellular space [51]. Lysosomal exocytosis involves the wholesale fusion of the lysosomal limiting membrane with the PM and release of the contents, including the intra-lysosomal vesicles (ILVs) and enzymes, whilst exosome release involves the fusion of the multi-vesicular body (MVB) with the PM, releasing small (approximately 50–150 nM diameter) vesicles into the extracellular space [87,88]. It seems likely that both are occurring in these disease models demonstrating that the consequences of lysosomal blockade are not confined to these endolysosomal organelles.

Two proteins identified in the WCP stand out as potentially very important in managing the lipid burden in these disease models: ABCA3 and the tetraspanin family member CD63. Both proteins are organized into membrane microdomains in the limiting membrane of endolysosomal compartments [89,90]. The lipid transporter ABCA3 is highly expressed in the brain but is well studied for its role in producing surfactant in the lungs [91–93]. Surfactant is a secreted lipid and protein mixture that coats the alveoli. Before secretion, surfactant components are packaged into lamellar bodies, a type of lysosome-related organelle (LRO) or specialized form of the MVB [94]. ABCA3 pumps lipids and cholesterol into lamellar bodies producing characteristic lipid whorls that are morphologically similar to those seen here (Fig 1F) and in patient and animal models of sphingolipidoses. Importantly, ABCA3 can transfer a broad variety of lipids including phospholipids, cholesterol and the sphingolipids sphingomyelin and glucosylceramide [41]. This suggests that the increased abundance of ABCA3 observed in the HEXA-deficient cell lines may contribute to the formation of the multilamellar membrane structures seen in these cells. In addition to this packaging role, ABCA3 regulates the release of lamellar bodies via fusion with the PM: deletion of ABCA3 drastically reduces exosome release, whilst overexpression enhances release [47]. Therefore, it is possible that the increased abundance of ABCA3 observed in these diseased neurons is contributing to both lipid whorl formation and transport of GM2 between the endolysosome and PM. Interestingly, SCARB2, NPC1 and NPC2 are also found on lamellar bodies and function to translocate cholesterol and sphingolipids between endolysosomes and other membranous compartments, particularly the ER [95–98]. Thus, formation of lamellar-like bodies, similar to those in lung cells and seen in Neimann Pick disease type C [99], may be a mechanism by which storage and subsequent release occurs in the GM2 gangliosidoses.

Four tetraspanin proteins (TSPAN3/6/7 and CD63) were significantly upregulated in the WCP of HEXA deficient neurons. These are important scaffolding proteins for sorting and loading cargoes in MVBs and are enriched on exosomes and play important roles in exosome formation [48,49,100,101]. CD63 sorts cholesterol into ILVs to be used as a storage molecule or to be released via exosomes. NPC1/2 can then retrieve this cholesterol in either the cell-of-origin or after the exosome has been taken up by another cell [102]. The packaging and release of cholesterol in this manner is thought to be a mechanism by which exosomes are used to remove excess cholesterol in Neimann Pick disease type C where cholesterol release in exosomes is increased [102]. Although CD63 is the best studied of these TSPANs, recently TSPAN3 was shown to be more efficient at sorting extracellular vesicles than CD63 [49]. Thus, the increased abundance of these tetraspanin proteins, seen in these neuronal disease models supports a significant role for exosome release in GM2 gangliosidoses.

The proteomic profile of these cells paints a picture of specific dysfunction of lipid processing and an activation of alternative pathways to evacuate pathological GM2 to compensate for the lack of functional HexA. The relevance of these mechanisms to disgorge excess GM2 is supported by the lysosomal limiting membrane proteins in the PMP data and the specific and significant increase in abundance of GM2 seen in the PM glycan profiling. This showed continuing accumulation of GM2 from 14 dpi increasing 3-fold by 28 dpi as the cells exocytose their defective endomembrane compartments. Whilst the majority of exocytosed GM2 may be released from the cell, EM images (Fig 1F) demonstrate that some of the lamellar inclusions are contiguous with the limiting membrane thereby facilitating their incorporation into the PM. This increase in GM2 at the PM is likely to be playing a specific role in disruption of membrane microdomains further contributing to the altered PM protein composition. There are several PM proteins that are altered in their abundance, both increased and decreased, that are not lysosomal proteins and so have been disrupted in their trafficking via pathways other than lysosomal exocytosis. These changes are likely driven by altered endo- and exocytosis of PM components due to the crucial role GSLs play in these recycling pathways [6,103]. In support of defective trafficking altering the PM proteome, several synaptic proteins (NTKR2, CACNG8, CNTNAP4, NRP1) were quantified in the WCP as not significantly changed but were changed in abundance at the PM, supporting a trafficking blockade with consequences for synaptic function.

To explore whether lysosomal exocytosis is a conserved mechanism in other gangliosidoses, a model of GM1 gangliosidosis was generated. The WCP at 28 dpi for both gangliosidosis models revealed several shared features in these disease models with significant accumulation of several lysosomal proteins (Fig 5C and 5D). A recent paper exploring the proteomic changes in microglia with another LSD, Batten disease caused by loss of CLN3, identify similar changes to those seen here, including increased TTYH3, PTTG1IP, LAMTOR3, PSAP, STARD3NL, TSPAN7, TMEM192 and NPC1 [86]. The similarities in these WCP datasets may indicate which lysosomal changes are shared across LSDs. In the gangliosidosis models here, in addition to increases in lysosomal proteins, there was increased abundance of proteins involved in lysosomal trafficking including the Arf-like GTPase ARL8a. This protein is localized to lysosomes and redistributes lysosomes to the cell periphery suggesting it may be contributing to lysosomal exocytosis (discussed further below) [104,105]. Analysis of the PMP data for the two gangliosidosis models revealed significant increases in lysosomal limiting membrane proteins versus SCRM control cells (Fig 5E). This supports that lysosomal exocytosis is a shared feature of GM1 and GM2 gangliosidosis and may be a more general mechanism seen in LSDs. Whether the recycling and endocytosis of these limiting membrane proteins is particularly disrupted in lipid-associated LSDs versus non-lipid LSDs is an interesting question for future studies. From these data there were very few other shared changes at the PM with the most significant increase being the PM amino-acid transporter SLC38A2 and the most decreased protein being an inactive carbonic anhydrase CA10.

The trafficking defects and therefore the proteomic changes at the PM may be specific to the type of ganglioside that is accumulating in these cells. Unfortunately, the PMP datasets for GM1 and GM2 disease models had relatively few proteins confidently quantified limiting our ability to reliably identify proteins that were differently changed in abundance

between these two disease models. Direct interactions between membrane proteins and ganglioside headgroups are difficult to define but there is growing evidence that some of these interactions are specific and may help explain the different phenotypes seen in different GSL-related diseases [106]. Recently, the human ganglioside interactome for GM3, GM1 and GD1a was explored using photo-clickable GSLs [40]. This work identified membrane proteins that specifically bind to the different ganglioside-headgroups of these lipids. Interestingly, many of the proteins identified in our proteomics data including lysosomal (TSPAN3, CD63 and LAMP1) and non-lysosomal proteins (TTYH3, NTRK2/3, ROR1, EGFR amongst others) are identified in this study to be potential direct ganglioside binders. GM2 is a precursor to higher order gangliosides and is not normally present at high abundance in the neuronal PM, thus the increase in amount seen here may disrupt the formation of correct ganglioside-protein interactions. The overlapping list of proteins identified here suggests that the excess GM2 may be disrupting normal GSL-protein interactions. The excessive amounts of inappropriately localized GM2 at the PM could compete with other GSLs to form non-native complexes disrupting normal trafficking including altering the endocytosis of PM proteins. Several ganglioside binding proteins have been shown to bind a range of gangliosides with different affinities and in this way interaction partners identified for GM3, GM1 and GD1a may be "hijacked" by excess GM2 [70,107]. Future work probing these specific interactions in vitro as well as more sensitive PM mass spectrometry to confidently identify the differences between these models would help highlight whether direct interactions between GSLs and membrane proteins drive distinct changes at the PM.

The abundance changes at the PM for synaptic proteins involved in calcium signalling, synaptic vesicle recycling and neurotransmitter binding have clear consequences on the electrical activity of disease neurons. The ΔHEXA-1 neurons exhibit increased excitation both spontaneously and with extrinsic electrical stimulation. Synaptic alterations are well documented in LSDs but the molecular mechanisms driving this remain unclear. The work presented here supports that rather than a general neuronal dysfunction, there are specific changes to the composition of the PM that in this neuronal type, cortical glutamatergic neurons, results in hyperactivity. Synaptic vesicle recycling shares many characteristics with the endolysosomal system including shared proteins involved in the fusion, trafficking and sorting of these compartments [108]. Whilst previously considered discrete pathways, evidence now suggests these two pathways form a continuum and that endolysosomal proteins can regulate SV recycling. An example of this joint pathway converges on ARL8a, an adaptor protein that aids in shuttling lysosomes and synaptic vesicle precursors along microtubules towards synapses [109,110]. Increases in the abundance of ARL8a in both ΔHEXA and ΔGLB1 cells may be driving anterograde movement of defective lysosomes away from the soma for eventual exocytosis. Due to the shared machinery involved, it may also be shuttling synaptic vesicles to synapses, providing more synaptic material that may be directly responsible for the greater excitatory response seen in ΔHEXA i3Ns. These observations are consistent with work demonstrating that overexpression of ARL8 facilitated increased evoked neurotransmission in mouse hippocampal neurons [111].

Several studies using different cellular and animal models of LSDs have identified synaptic signalling defects with a range of different consequences depending on the specific disease and system studied [112,113]. In several cases reduced signalling is observed, caused by fewer synaptic vesicles, loss of key synaptic proteins and loss of dendritic spines. In a recent study developing iPSC models from Tay–Sachs patient fibroblasts, exocytotic activity was reduced in response to chemical stimuli [26]. However, other work from Niemann Pick Disease type C mice, display increased synaptic transmission potentially driving chronic excitotoxicity and neurodegeneration [114,115]. Enhanced exocytosis has also been observed in a mouse model of mucolipidosis IV supporting a role for glutamate neurotoxicity and synaptic exhaustion in neuronal pathology [116]. iPSC neuronal lines derived from different Niemann Pick Disease type C patients have been shown to possess a higher resting voltage than healthy neurons and are easier to depolarize [117]. Niemann Pick Disease type C neurons accumulate not only cholesterol but also multiple sphingolipids including GM2 [118,119]. A similar change is seen in the ΔHEXA-1 neurons here, where cells are more readily depolarized with electrical stimulation. Disruptions in synaptic function are challenging to study, but the system used here clearly demonstrates that GM2 accumulation in these neurons alters the PM composition resulting in greater electrical excitation both spontaneously and with stimulation.

Enhancement of lysosomal exocytosis has been explored as a method to ameliorate the effects of LSDs via chemical induction, using cyclodextrin, or induced TFEB expression [120,121]. This approach has been used to reduce the accumulation of cholesterol, GM2 and GM3 in Niemann Pick Disease type C models and GM2 in GM2 gangliosidosis models [27,122,123]. Although enhancing lysosomal exocytosis could effectively reduce the accumulation of storage materials, the deleterious consequences of altered PM lipid and protein composition suggest caution should be used in implementing this approach [124]. Furthermore, in other neurodegenerative diseases, the release of exosomes is known to pathologically "seed" bystander cells by passing misfolded toxic amyloid β and tau proteins to nearby cells [125]. In GM2 gangliosidosis, exosomes or lysosomal exocytosis may pass GM2 to nearby cells such as other neurons, astrocytes and microglia, with these cell types also unable to degrade GM2, leading to its accumulation in these bystander cells. Beyond this potential impact of exosome uptake, microglia and astrocytes are both phagocytic cells [126,127] and as they clear up neuronal cell debris they will accumulate GM2 driving further dysfunction in these cell types and contributing to pathological phenotypes.

The data presented here provide deep molecular insights into the mechanisms driving GM2 gangliosidosis pathology. We define the specific changes occurring within the endolysosomal compartments, the impact this has on the PM composition and the subsequent consequences for electrical signalling within the neuronal network. We also demonstrate several shared phenotypes such as accumulation of lysosomal proteins at the PM in a model of GM1 gangliosidosis. The i3N cell lines developed here represent robust, reproducible and versatile models for the study of gangliosidoses enabling broad, unbiased analysis of protein and lipid composition of different cellular compartments. These models represent an excellent resource for future research including drug screening to alleviate or reverse lysosomal and synaptic phenotypes. These models can be further expanded to explore the changes that occur in different gangliosidoses and more distantly related LSDs. This work clearly identifies that gangliosidoses are not just lysosomal storage diseases but also involve complex plasma membrane changes that directly affect the electrical activity of neurons.

## Methods

### Cell line maintenance

Human CRISPRi-i3N stem cells [28] containing a stably integrated dead Cas9 (dCas9) were maintained at 37 °C and 5% $CO_2$ with either daily media changes of complete E8 medium or triweekly media changes with complete E8 Flex medium, on plates coated with Matrigel diluted 1:50 in DMEM/F-12 HEPES. Cells were split using either 0.5 mM EDTA to detach cells, where they were replated as colonies in E8 medium, or Accutase, where they were replated as individual cells with E8 medium plus 10 µM Rock Inhibitor Y-27632 (Tocris).

Differentiation of stem cells into neurons was done as previously described, with slight modifications [28]. Initial differentiation was induced following Accutase treatment and seeding of $6–8 \times 10^6$ stem cells into a 10 cm Matrigel-coated dish using induction media (IM), comprised of DMEM/F12 HEPES, supplemented with 1× N2 supplement, 1× NEAA, 1× Glutamax and 2 µg/ml Doxycycline. During initial plating, the media was supplemented with 10 µM Rock Inhibitor and media was changed daily for three days.

Partially differentiated neurons were detached with Accutase and replated onto plates coated with 100 µg/ml poly-l-ornithine (PLO), in cortical neuron (CN) media comprised of Neurobasal Plus medium supplemented with 1× B27 supplement, 10 ng/ml NT-3 (Peprotech), 10 ng/ml BDNF (Peprotech) and 1 µg/ml laminin. During initial plating, media was supplemented with 1 µg/ml doxycycline. Media was half changed twice or thrice weekly until day 14 unless otherwise specified.

### Generation of CRISPR knockdown cell lines

Guide RNAs targeting the *HEXA*, *HEXB* or *GLB1* genes were designed using the CRISPick server (Broad Institute, S1 Table) to target either the start of or immediately prior to the first exon and complimentary overhanging primer dimers were cloned into the BpiI restriction site on the pKLV plasmid (gift from Evan Reid).

For lentiviral production, HEK293T cells were maintained in DMEM supplemented with 10% FBS at 37 °C and 5% $CO_2$ and were passaged twice weekly with Trypsin-EDTA. Two × $10^6$ cells were transfected using 8 µl of Trans-IT and 2 µg of a 3:2:1 ratio of CRISPRi pKLV vector:pCMVΔ8.91:pMD.G lentiviral packaging vectors in Optimem. The transfection mix was added to cells dropwise and DMEM was replaced with E8 media after 24 h. After 48 h, virus containing E8 media was sterile filtered, diluted 1:1 with Fresh E8 and was frozen at −70 °C.

Five × $10^5$ i3N stem cells were transduced with 2 ml of 1:1 E8 media/lentivirus-containing media supplemented with 10 µM Rock Inhibitor and 10 µg/ml Polybrene and spun onto Matrigel-coated 6-well plates at 750 × $g$ for 1 h at 32 °C, then incubated at 37 °C. After 24 h, media was changed with fresh E8 media and then 48 h post transduction, media was changed with E8 medium supplemented with 1 µg/ml puromycin, daily for two days. Selected cells were subsequently grown without puromycin.

### Quantitative PCR (qPCR)

RNA was extracted from 2 × $10^6$ cells as per manufacturer's instructions using a Purelink RNA extraction kit (Invitrogen). RNA was reverse transcribed to cDNA using a High-Capacity RNA-to-cDNA kit (Applied Biosystems, Thermo Scientific), as per manufacturer's instructions. RT-qPCR was performed using TaqMan Fast Advanced Mastermix (Applied Biosystems, Thermo Scientific), 50 ng cDNA and FAM-labelled Taqman Gene Expression Assay probes (Applied Biosystems, Thermo Scientific) specific to each gene (S6 Table). DNA amplification was performed using a CFX96 Real Time System (Bio-Rad) using 40 cycles of melting at 95 °C for 15 s, annealing and extension at 60 °C for 1 min. Experiments were conducted with $n = 3$ technical and $N = 3$ biological triplicates. Gene expression changes were calculated using the ΔΔCt method [128] and using GAPDH as control. Statistical significance of fold changes were determined using the one-way ANOVA statistical test in GraphPad Prism 9.

### Enzyme activity assays

HEXA activity assays were modified from previously published assays [32] as follows. Cells were lysed in 10 mM Tris pH 7.4, 1% Triton X-100, 150 mM NaCl, 5 mM EDTA, supplemented with 1 cOmplete protease inhibitor tablet (Sigma Aldrich) per 50 ml buffer and were centrifuged (12,000 × $g$, 10 min) to remove debris. Volumes of cleared lysates were normalized following Pierce BCA assay (Thermo Fischer) and diluted to 40 µg/ml in PBS with 0.5% BSA. Samples were divided in two, with one half heated to 51.5 °C for one hour. The 4-Methylumbelliferyl-2-Acetamido-2-Deoxy-β-D-Glucopyranoside (MUG, Sigma–Aldrich) or 4-Methylumbelliferyl-6-sulfo-N-acetyl-β-D-glucosaminide (MUGS, Sigma Aldrich) substrates were dissolved in 0.2 M $Na_2HPO_4$, pH 4.4 to a concentration of 3 mM. Heated and unheated samples were added to MUG substrate and unheated samples were added to MUGS substrate at a ratio of 1:2 v:v sample:substrate ratio. Lysate/substrate mixes were incubated at 37 °C for 30 min at 200 RPM in the dark before quenching with a 1:12 (v:v) sample:stopping buffer (250 mM glycine carbonate, pH 10.0) ratio. Fluorescence was read on a Clariostar plate reader (BMG LabTech) with excitation and emission wavelengths of 360 and 450 nm, respectively. Experiments were conducted with technical and biological triplicates and statistical significance of differences in activity levels were determined using the one-way ANOVA statistical test in GraphPad Prism 9.

### Quantification of gangliosides by HPLC

Glycosphingolipids were analyzed as previously described [37,129]. Briefly, lipids were extracted from samples of 2 × $10^6$ cells using 4:8:3 Chloroform:Methanol:PBS and further purified using solid phase C18 columns (Telos, Kinesis). Fractions were eluted twice with 1 ml of 98:2 Chloroform:methanol, and twice with 1 ml of 1:3 Chloroform:methanol and 1 ml Methanol, then dried down under a nitrogen stream and digested overnight using recombinant endoglycoceramidase I (a gift from David Priestman). Cleaved glycans were labelled with 2-Anthranilic Acid (2AA) and purified using a DPA-6S

Solid Phase Extraction amide column (Supelco). The purified, 2AA-labelled glycans were then separated and quantified by normal-phase HPLC and areas under the curve for each peak were compared to 2.5 pmol of 2AA-labelled chitotriose standard (Ludger) for quantification. For comparison between samples, this quantification was corrected for the number of cells seeded. For glycan identification, peak elution times were compared to a pre-determined mix of commercially available gangliosides (S2 Fig). Separation was performed using a TSK gel-Amide 80 column (Anachem) in combination with a Waters Alliance 2,695 separations module and in-line Waters 474 fluorescence detector, set to excitation and emission wavelengths of 360 and 425 nm, respectively. All chromatography was performed at 30 °C and solvent composition, gradient conditions and run parameters are as described previously [129]. Experiments were performed in biological triplicate and significant differences in the quantity of each ganglioside species were determined using the one-way ANOVA statistical test in GraphPad Prism 9.

## Quantification of plasma membrane glycosphingolipids

Biotinylation of PM gangliosides was performed by modification of methods described previously for labelling of cell surface glycosylated proteins [130]. Briefly, $2 \times 10^6$ cells were washed twice in ice-cold PBS. Surface sialic acid residues were oxidized and biotinylated by incubation in the dark for 30 min with gentle rocking at 4 °C using an oxidation/biotinylation mix comprising 1 mM sodium meta-periodate, 100 µM aminooxy-biotin (Biotium , Hayward, CA) and 10 µM aniline (Sigma–Aldrich) in ice-cold PBS, pH 6.7. The reaction was quenched with 1 mM glycerol and cells were washed twice in ice-cold PBS. Labelled ganglioside headgroups were then purified and quantified using methods as described above for whole-cell analysis. Attempts to enrich and separate biotinylated gangliosides or cleaved glycan headgroups using streptavidin enrichment were unsuccessful. However, due to the altered elution profile of biotinylated glycan headgroups specific subsets of gangliosides were able to be quantified within the lipid mixture (S6 Fig). Identification of specific biotinylated glycan headgroups was determined by in vitro labelling of pure lipid species in liposomes composed of 20% ganglioside, 58% phosphatidylcholine, 20% cholesterol and 2% rhodamine-PE (S7 Table).

## Transmission electron microscopy

Cell monolayers were fixed in 50% cell culture medium and 50% EM fixative (2% (w/v) paraformaldehyde and 2.5% (v/v) glutaraldehyde in 100 mM sodium cacodylate buffer, pH 7.2 at 37 °C and this medium was immediately replaced with warm 100% fixative for 1 h. Cells were washed with 100 mM sodium cacodylate buffer and post-fixed with 1% $OsO_4$ in 100 mM sodium cacodylate buffer, pH 7.2 for 1 h at room temperature. They were then washed with sodium cacodylate buffer followed by 50 mM sodium maleate buffer (pH 5.2) and 'en bloc' stained with 0.5% (w/v) uranyl acetate in 50 mM sodium maleate buffer, pH 5.2. The cells were sequentially dehydrated using 50% through to 100% ethanol and embedded in 50:50 v:v ethanol:agar 100 resin (Agar Scientific) and finally 100% Agar 100 resin for 24 h. A resin-filled BEEM capsule was then inverted over the cell monolayer and the resin was polymerized in an embedding oven for 24 h at 60 °C. The BEEM capsules were removed by immersion in liquid nitrogen.

Ultrathin sections were cut 'en face' to the plane of the monolayer using a diamond knife mounted on a Leica Ultracut UC7 ultramicrotome (Leica, Milton Keynes, UK), transferred to EM grids and stained with uranyl acetate and Reynold's lead citrate. The sections were analyzed using a Tecnai G2 Spirit BioTWIN transmission electron microscope (FEI) at an operating voltage of 80 kV and images were recorded using a 4 Megapixel Gatan US1000 CCD camera.

Twenty images per cell line were captured at 13,000× magnification. Images were analyzed with ImageJ software, using a scale bar to define size with the analyze > set measurements function. The freehand selection tool and analyze > measure functions were then used to quantify endolysosome numbers and determine 2D endolysosome area measurements. Data were analyzed by one-way ANOVA using GraphPad Prism 9.

## Immunofluorescence microscopy

Partially differentiated 3 dpi neurons were seeded onto coverslips in 24-well plates. Prior to seeding, coverslips were acid etched with 100% acetic acid and tumbling for 24 h, then washed with ethanol and coated with 100 μg/ml PLO for 24 h. At 14 dpi, coverslips were fixed with Cytoskeletal Fixing Buffer (300 mM NaCl, 10 mM EDTA, 10 mM Glucose, 10 mM $MgCl_2$, 20 mM PIPES, pH 6.8, 2% Sucrose, 4% paraformaldehyde) for 10 min at RT. Coverslips were permeabilized with 0.1% Saponin in PBS for 10 min followed by blocking in 0.01% saponin, 1% BSA in PBS, and stained in the same buffer, with specific antibodies described below. Coverslips were stained for 1 h in primary antibody, followed by three washes in 0.01% saponin in PBS, following by staining with secondary antibody for 45 min. Coverslips were then washed three times in 0.01% saponin in PBS and mounted in Prolong Gold Antifade with DAPI (Thermo Fisher). Antibodies used were rabbit polyclonal anti-LAMP1 (Abcam, ab24170), mouse monoclonal anti-GM2 (gift from Kostantin Dobrenis), AF555 donkey anti-rabbit (Invitrogen, A31572) and AF488 goat anti-mouse IgM (Invitrogen, A21042). Two coverslips per cell line were stained and imaged to produce 15 images. This was then repeated in triplicate with cells grown on different days.

All images were collected on a Zeiss LSM 780 laser scanning microscope using Zen Black software and a 63× oil immersion lens and 512 × 512 pixel size images in a 16-bit range were obtained. Automated analysis and object-based colocalization of image sets was performed using CellProfiler software [131]. The methodology is based on the CellProfiler online tutorials (https://tutorials.cellprofiler.org/) and the Cell Profiler Workflow is included as a Supplementary Data File (https://doi.org/10.17863/CAM.118836). Briefly, images were split into color layers with nuclei identified on the blue layer as objects between 25 and 80 pixels in diameter. Objects with diameters of 2–20 pixels on the red layer (LAMP1 positive) were identified as endo/lysosomes and on the green layer as GM2. Colocalization was determined by identifying which green objects overlay with red. Counts for Nuclei, GM2, LAMP1 and GM2 positive for LAMP1 were used on a per image basis to produce GM2/cell and GM2 in lysosomes/Cell variables.

## Calcium activity assay

Partially differentiated 3 dpi neurons were seeded in 100 μl of media into PLO-coated 96-well plates (Corning, 3595) at a density of $5 \times 10^4$/well. Twenty-four hours after seeding, 100 μl of diluted (1:100 v:v in CN media) Neuroburst Orange Lentivirus (Sartorius) expressing a genetically-encoded fluorescent calcium sensor was added to each well. Twenty-four hours after virus addition, all media was removed and replaced with 200 μl of fresh CN media. Wells were imaged daily using an Incucyte S3 Live-Cell Analysis System (Sartorius) in movie mode acquisition for 2 min/well with 3 images/second, with a 4× objective lens and 400 ms acquisition time, up to 30 dpi. Analysis of the recorded movies was performed using the Neuronal Activity Analysis Software Module [74]. Briefly, active objects (cells/cell clusters) that burst above a minimum threshold were compared to every other object in the image to generate a correlation value between −1 and 1 with 1 being highly synchronous indicating high network connectivity. The following analysis parameters were chosen: object size 30 μm; minimum cell width 6 μm; sensitivity −3; edge split off; minimum burst intensity 0.2. Correlation scores for technical quadruplicates for individual cell lines were averaged and then these means for three cell lines; SCRM, ΔHEXA-1 and ΔHEXB-1 were combined on a per experiment basis and presented to identify the time points where cells were signalling in high synchronicity across two separate experiments.

## Proteomics

**Proteomics—cell culture.** Cell lines for proteomic analysis were plated in triplicate to yield three independent biological replicates. Partially differentiated 3 dpi i³Ns lines of SCRM, ΔHEXA-1, ΔHEXA-2, ΔHEXB-1 and ΔHEXB-2 cells were seeded at $2 \times 10^6$/well in PLO coated plates for Whole-Cell Proteomics (WCP) and at $10 \times 10^6$ cells/plate in PLO coated 10 cm plates for Plasma Membrane Proteomics (PMP). Cells were left to differentiate until their 14 or 28 dpi time point with bi-weekly half media changes.

**Sample preparation for WCP.** Cells were washed three times with PBS and scaped into low-bind Eppendorf tubes. Cells were pelleted at 500 × g for 10 min, the PBS removed and snap frozen on LN2 and stored at −80 °C until all samples were ready for simultaneous processing. Cell pellets were resuspended in 50 µL resuspension buffer (76 mM HEPES pH 7.55, 6 mM MgCl$_2$, Benzonase (1,400 U/mL) and 15 mM TCEP) by pipetting. Then, 18.75 µL of 20% lithium dodecyl sulfate was immediately added to the cell suspension using a low retention pipette tip (RPT, StarLab) and pipetted to mix. Nucleic acids were fragmented by 30 s on/30 s off sonication in a Bioruptor Pico sonicator (Diagenode) for 10 min at 4 °C. Samples were then incubated for 15 min at 37 °C to ensure complete reduction. Samples were alkylated by adding 6 µL of 187.5 mM methyl methanethiosulfonate (final concentration 15 mM) and incubating at room temperature for 15 min. An amount of 5 µL aliquots of each sample were diluted 2× in water and compared to a standard curve of BSA in the same buffer using a reducing agent-compatible BCA assay (Thermo Fisher). An amount of 25 µg of each sample was taken and the volumes of each lysate were equalized using resuspension buffer with 5% lithium dodecyl sulfate.

To each sample a 10% volume of 12% phosphoric acid was added to acidify samples to approximately pH 2, completing denaturation. 6× volumes of wash buffer (100 mM HEPES pH 7.1, 90% methanol) was then added and the resulting solution was loaded onto a S-trap (Protifi) using a positive pressure manifold (Tecan M10), adding not more than 150 µL of sample at a time (approximately 80 PSI). In-house fabricated adaptors were used to permit the use of S-traps with the manifold. Samples were then washed 4× with 150 µL wash buffer. To remove any remaining wash buffer S-traps were centrifuged at 4000 × g for 2 min. To each S-trap, 30 µL of digestion solution (50 mM HEPES pH 8, 0.1% sodium deoxycholate) containing 1 µg Trypsin/lysC mix (Promega) was added. S-Traps were then loosely capped and placed in low adhesion 1.5 mL microfuge tubes in a ThermoMixer C (Eppendorf) with a heated lid and incubated at 37 °C for 6 h. Where digestion was carried out overnight the Thermomixer was set to 4 °C after 6 h. Peptides were recovered by adding 40 µL digestion buffer to each trap and incubating at room temperature for 15 min before slowly eluting with positive pressure (2–3 PSI). Traps were subsequently eluted with 40 µL 0.2% formic acid and 40 µL 0.2% formic acid, 50% acetonitrile in the same manner. Eluted samples were then dried in a vacuum centrifuge equipped with a cold trap prior to TMT labelling.

**Sample preparation for PMP-MS.** Sialic acid moieties present on the extracellular side of PM proteins were oxidized to produce an aldehyde group using Sodium Periodate, followed by an analine-catalyzed oxime ligation to aminooxy-biotin. Cells prepared for PMP were washed three times with 5 ml of ice-cold PBS pH 7.4 and then incubated under 5 ml of biotinylation mix (1 mM Na Periodate, 100 µM aminooxy-biotin, 10 mM aniline in ice-cold PBS pH 6.7) and wrapped in foil with rocking, at 4 °C for 30 min. After biotin labelling, the reaction was quenched with 5 ml of 2 mM glycerol, the biotinylation/quenching reaction was then removed, and the cells washed 3× with PBS pH 7.4. All PBS was removed, and cells were scraped into a low bind Eppendorf in 500 µl of lysis buffer (10 mM Tris pH 7.4, 1% Triton X-100, 150 mM NaCl, 5 mM EDTA and 1 protein inhibitor tab per 50 ml).

Samples in lysis buffer were lysed through end over end rotation for 1.5 h at 4 °C. Lysates were clarified with centrifugation at 20,000 × g for 10 min. The supernatant was then removed to a fresh low-bind Eppendorf tube and snap frozen on LN2 and stored at −80 °C until all samples were ready for simultaneous enrichment steps. Frozen samples were thawed, and a BCA assay was performed. Samples were normalized with lysis buffer to the lowest concentration to ensure the same mass of protein went into subsequent enrichment steps. An amount of 50 µl of Neutravidin bead slurry per sample was washed with 1 ml of lysis buffer four times (including resuspension, centrifugation 500 × g for 5 min, removal of liquid, repeat). Normalized samples were then added to neutravidin beads and incubated with end over end rotation for 2.75 h at 4 °C.

Beads with enriched samples were washed using a vacuum manifold and SnapCap filter columns (Pierce). Bead/sample slurry was moved into the snapcap columns and allowed to drain. Each sample was then washed 20× with 400 µl of lysis buffer, followed by 20× washes with 0.5% SDS in PBS pH 7.4, followed by 10× washes with Urea buffer (6 M urea, 1 M TEAB (Thermo), pH 8.5). Columns were removed, capped to prevent liquid loss and the beads were incubated with 400 µl of reduction/alkylation solution (10 mM TCEP (Thermo), 20 mM Iodoacetamide, in Urea buffer), shaking at 850 rpm

for 30 min in the dark. Samples were returned to the vacuum manifold and the beads drained before a further 10× washed in urea buffer. Columns were again capped, and the beads resuspended in 400 µl of 50 mM TEAB and removed to a low adhesion Eppendorf tube. The columns were rinsed again 2× with 400 µl of 50 mM TEAB and the washes combined with the sample. Beads were pelleted gently at 500 × $g$ for 2 min and all supernatant was removed. Beads were then resuspended in 50 µl of 50 mM TEAB + 0.5 µg of trypsin (MS grade, Pierce) (trypsin stocks made up at 1 µg/µl in 50 mM acetic acid) and incubated at 850 rpm and 37 °C, overnight. The following day, the beads were pelleted at 500 × $g$ for 5 min and the supernatant removed and stored. The beads were washed with a further 40 µl of 50 mM TEAB and pelleted again, the wash supernatant was combined with the sample and then samples were dried in a vacuum centrifuge before storage at −20 °C awaiting TMT labelling.

**TMT labelling and clean-up.** Dried samples were resuspended in 21 µL 100 mM TEAB pH 8.5. After warming to room temperature, 0.5 µg TMTpro/0.2 µg TMT reagents (Thermo Fisher) were resuspended in 9 µL anhydrous acetonitrile which was added to the respective samples and incubated at room temperature for 1 h. A 3 µL aliquot of each sample was taken and pooled to check TMT labelling efficiency and equality of loading by liquid chromatography–mass spectrometry (LC–MS). Samples were stored at −80 °C in the interim. After checking each sample was at least 98% TMT labelled, total reporter ion intensities were used to normalize the pooling of the remaining samples such that the final pool should be as close to a 1:1 ratio of total peptide content between samples as possible. This final pool was then dried in a vacuum centrifuge. Whole cell proteomics samples were acidified to a final 0.1% trifluoracetic acid (approximately 200 µL volume) and formic acid was added until the sodium deoxycholate visibly precipitated. Amount of 4 volumes of ethyl acetate were then added and the sample vortexed vigorously for 10 s. The sample was then centrifuged at 15,000 × $g$ for 5 min at room temperature to effect phase separation. A gel loading pipette tip was used to withdraw the lower (aqueous) phase to a fresh low adhesion microfuge tube. If any obvious sodium deoxycholate contamination remained, the two-phase extraction with ethyl acetate was repeated. The sample was then partially dried in a vacuum centrifuge. Whole cell and PMP samples were brought up to a final volume of 1 mL with 0.1% trifluoracetic acid. Formic acid (FA) was added until the pH was <2, confirmed by spotting onto pH paper. The sample was then cleaned up by solid-phase extraction using a 50 mg tC18 SepPak cartridge (Waters) and a positive pressure manifold. The cartridge was wetted with 1 mL 100% Methanol followed by 1 mL acetonitrile, equilibrated with 1 mL 0.1% trifluoracetic acid and the sample was loaded slowly. The sample was passed twice over the cartridge. The cartridge was washed 3× with 1 mL 0.1% trifluoracetic acid before eluting sequentially with 250 µL 40% acetonitrile, 70% acetonitrile and 80% acetonitrile and dried in a vacuum centrifuge.

**Mass spectrometry.** Mass spectrometry data was acquired using an Orbitrap Lumos as previously described [132]. An Ultimate 3000 RSLC nano UHPLC equipped with a 300 µm ID × 5 mm Acclaim PepMap µ-Precolumn (Thermo Fisher Scientific) and a 75 µm ID × 50 cm 2.1 µm particle Acclaim PepMap RSLC analytical column was used. Loading solvent was 0.1% FA, analytical solvent A: 0.1% FA and B: 80% MeCN + 0.1% FA. All separations were carried out at 40 °C. Samples were loaded at 5 µL/min for 5 min in loading solvent before beginning the analytical gradient. The following gradients were used: 3%–7% B over 4 min, 7%–37% B over 173 min, followed by a 4-min wash at 95% B and equilibration at 3% B for 15 min or 3%–7% B over 3 min, 7%–37% B over 116 min, followed by a 4-min wash at 95% B and equilibration at 3% B for 15 min. Each analysis used a MultiNotch MS3-based TMT method [133]. The following settings were used: MS1: 380–1,500 Th, 120,000 Resolution, $2 \times 10^5$ automatic gain control (AGC) target, 50 ms maximum injection time. MS2: Quadrupole isolation at an isolation width of m/z 0.7, CID fragmentation (normalized collision energy (NCE) 34) with ion trap scanning in turbo mode with $1.5 \times 10^4$ AGC target and 120 ms maximum injection time. MS3: In Synchronous Precursor Selection mode, the top 10 MS2 ions were selected for HCD fragmentation (NCE 45) and scanned in the Orbitrap at 60,000 resolution with an AGC target of $1 \times 10^5$ and a maximum accumulation time of 150 ms. Ions were not accumulated for all parallelizable time. The entire MS/MS/MS cycle had a target time of 3 s. The mass spectrometry data have been deposited to the ProteomeXchange Consortium via the PRIDE [134] partner repository.

**Analysis of mass spectrometry data.** Data was processed in Proteome Discoverer 2.2 using the Mascot search engine and a Uniprot human database (downloaded 11/01/2021). The results were imported into Perseus [135] as a tab delimited text file. In Perseus the grouped reporter abundances were $\log_2$ transformed, data filtered to a minimum of 3 valid values and missing values replaced from a normal distribution. The replicates were grouped, and volcano plots generated, with significantly changed proteins determined using a two-sample $t$ test with an FDR cutoff of <0.05. Differences in protein abundance and adjusted $P$-value were used to ascribe significance and to generate volcano plots in GraphPad Prism 9.

The Database for Annotation, Visualization and Integrated Discovery (DAVID) was used for functional annotation and enrichment analysis of proteomic datasets [136]. Proteins that were significantly changed ($p < 0.05$) in abundance by >20% compared to SCRM controls and with ≥2 PSMs were searched against a background reference list of all proteins quantified in the respective dataset with ≥2 PSMs. For functional enrichment analysis of WCP, only proteins increased in abundance were selected for analysis against a background list as defined above. Cellular Component GO_Direct terms were then selected based on FDR.

**Multielectrode array measurements.** Wells of a 24-well MEA plate (Axion Biosystems) were coated with 100 μl of 100 μg/ml PLO and plates were incubated at room temperature overnight. Wells were washed three times with sterile water before being left to dry for approximately 1 h. A $4 \times 10^5$ neurons/well were seeded in 100 μl of CN Media/maintenance media + 1 μg/ml doxycycline and maintained as described above in 200 μl of CN media. A Maestro MEA EDGE system and the control software Navigator (Axion BioSystems) were used for signal recording and stimulation. Spike detection was performed with a threshold of 6 times standard deviations. Electrical activity of neurons was monitored regularly for activity and synchronicity. When recordings took place on the same day as the half-media changes, neurons were recorded prior to changing the media as changes temporarily alter neuronal activity. On the recording day, the plate was loaded into the MEA reader, equilibrated for 15 min at 37 °C and 5% $CO_2$. For spontaneous signalling measurements, recording was performed via AxIS for 3 min (200 Hz–3 kHz). To electrically stimulate the culture, biphasic voltage pulses defined in the Navigator software ('Neural Stimulation' setting, amplitude = 150–300 mV, duration = 200–400 μs) were delivered to the target electrodes at 0.1 Hz for 5–10 min. Recorded data were processed with the Navigator software and then subsequently using GraphPad Prism 9 software. Data collection, analysis and statistical analyses were carried out as recommended in [75,76].

## Supporting information

**S1 Fig. qPCR analysis of gene expression for the stem cell marker NANOG and neuronal markers MAP2 and β3-tubulin in SCRM and ΔHEXA cell lines at 0 and 14 dpi.** Fold change is calculated relative to 14 dpi SCRM controls, $N = 3$ biological replicates for NANOG and MAP2 and $N = 2$ biological replicates for β3-Tubulin were carried out in technical triplicate $n = 3$ (squares, triangles, circles) and the mean is displayed (grey lines). Significance was determined with a two-way ANOVA, ****$p \leq 0.0001$. Underlying data used to generate these figures are available in S1 Data at https://doi.org/10.17863/CAM.118836.
(TIF)

**S2 Fig. Identification and quantification of glycosphingolipid headgroups using HPLC analysis of cleaved and labelled glycan headgroups.** Elution profile of known GSL standards as reference for peak identification. Each glycan headgroup has a distinct elution volume (labelled).
(TIF)

**S3 Fig. Quantitative mass spectrometry from whole cell samples of ΔHEXA compared to ΔHEXB cell lines.** A volcano plot is shown with the horizontal axis showing average fold change across three biological replicates and the

vertical axis showing significance (two-sided *t* test) across the three replicates. Significance cutoffs of >20% change and *p*-value < 0.05 are indicated (red dotted lines). Beyond the subunits that have been knocked down there are only 3 proteins with significant changes in protein abundance between these cell lines. Underlying data used to generate these figures are available in S1 Data at https://doi.org/10.17863/CAM.118836.
(TIF)

**S4 Fig. qPCR analysis of a selection of lysosomal proteins involved in the CLEAR response including two that are increased in abundance in WCP data.** Gene expression levels are shown at 14 dpi for SCRM and ΔHEXA lines. Fold change is calculated relative to SCRM controls, *N* = 3 biological replicates were carried out in technical triplicate *n* = 3 (squares, triangles, circles) and the mean is displayed (grey line). Significance was determined with a one-way ANOVA, no significant (ns) differences are seen between SCRM and ΔHEXA lines. Underlying data used to generate these figures are available in S1 Data at https://doi.org/10.17863/CAM.118836.
(TIF)

**S5 Fig. qPCR analysis of a selection of high confidence targets identified as changed in abundance in PMP data at 14 dpi.** Fold change is calculated relative to SCRM controls, *N* = 3 biological replicates were carried out in technical triplicate *n* = 3 (squares, triangles, circles) and the mean is displayed (red line). Significance was carried out using a one-way ANOVA. Although some changes were determined to be significant in these data, the fold change was less than 2-fold meaning it did not satisfy the criteria for a reliable fold change by qPCR [137]. Underlying data used to generate these figures are available in S1 Data at https://doi.org/10.17863/CAM.118836.
(TIF)

**S6 Fig. HPLC elution profiles of biotinylated ganglioside headgroups.** A. Standards from individually labelled lipid species GT1b, GM1a and GM2. B. Ganglioside headgroup elution profiles of 28 dpi whole cell samples after surface labelling with aminoxybiotin. Underlying data used to generate these figures are available in S1 Data at https://doi.org/10.17863/CAM.118836.
(TIF)

**S1 Table. Guide RNA target sequences.**
(DOCX)

**S2 Table. High confidence targets identified in WCP of ΔHEXA and ΔHEXB compared with SCRM control cells at 14 dpi.**
(DOCX)

**S3 Table. High confidence targets identified in PMP-MS of ΔHEXA and ΔHEXB compared with SCRM control cells at 14 dpi.**
(DOCX)

**S4 Table. Targets identified in WCP-MS of ΔHEXA and ΔGLB1 compared with SCRM control cells at 28 dpi.** Data for individual and combined comparisons are shown with those that satisfy the criteria of significant (<0.05) abundance change of >1.2 fold change (FC) are colored blue.
(DOCX)

**S5 Table. High confidence targets identified in PMP-MS of ΔHEXA and ΔGLB1 compared with SCRM control cells at 28 dpi.**
(DOCX)

**S6 Table. TaqMan gene expression assay probe sets.**
(DOCX)

**S7 Table. Purified lipids used as standards for PM glycan profiling.**
(DOCX)

**S1 Data. All individual numerical values corresponding to data displayed in** Figs 1C, 1D, 1E, 1F, 2A, 2B, 2C, 2D, 2E, 3A, 3B, 4A, 4B, 4D, 5A, 5C, 5D, 5F, **S1, S3, S4, S5**.
(XLSX)

## Author contributions

**Conceptualization:** Alex S. Nicholson, Janet E. Deane.

**Data curation:** Robin Antrobus.

**Formal analysis:** Alex S. Nicholson, Robin Antrobus.

**Funding acquisition:** Janet E. Deane.

**Investigation:** Alex S. Nicholson, David A. Priestman, James C. Williamson, Reuben Bush, Henry G. Barrow, Emily Smith, Nicholas A. Bright.

**Methodology:** Alex S. Nicholson, David A. Priestman, Shannon J. McKie, Nicholas A. Bright, Janet E. Deane.

**Project administration:** Janet E. Deane.

**Resources:** Kostantin Dobrenis.

**Supervision:** Frances M. Platt, Janet E. Deane.

**Visualization:** Alex S. Nicholson, Janet E. Deane.

**Writing – original draft:** Alex S. Nicholson, Janet E. Deane.

**Writing – review & editing:** Alex S. Nicholson, David A. Priestman, Robin Antrobus, James C. Williamson, Shannon J. McKie, Nicholas A. Bright, Frances M. Platt, Janet E. Deane.

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
