## [Editor Report · Decision Letter 0]

Dear Dr Deane,

Thank you for submitting your manuscript entitled "Plasma Membrane Remodelling in GM2 Gangliosidoses Drives Synaptic Dysfunction" for consideration as a Research Article by PLOS Biology. We understand that your study has been revised in response to reviews from Review Commons. We have now had a chance to discuss your study, the reviews, and your response to reviewers with an Academic Editor with relevant expertise, and I am writing to let you know that we would like to send your submission back to the original reviewers for their input on the revision. 

Once your full submission is complete, your paper will undergo a series of checks in preparation for peer review. After your manuscript has passed the checks it will be sent out for review. To provide the metadata for your submission, please Login to Editorial Manager (https://www.editorialmanager.com/pbiology) within two working days, i.e. by Apr 28 2025 11:59PM.

Kind regards,

Luke

Lucas Smith, Ph.D.

Senior Editor

PLOS Biology

lsmith@plos.org

---

## [Decision Letter · Decision Letter 1]

Dear Dr Deane,

Thank you for your patience while we considered your revised manuscript "Plasma Membrane Remodelling in GM2 Gangliosidoses Drives Synaptic Dysfunction" for publication as a Research Article at PLOS Biology. This revised version of your manuscript has been evaluated by the PLOS Biology editors and two of the original reviewers from Review Commons. Please note that we asked reviewer 1 to also act as the Academic Editor for your paper. 

As you will see, below, both reviewers have suggested that we accept your study. However, before we can accept your paper, we would like to invite you to address a number of editorial, data, and other policy-related requests in a last, short revision. These requests are detailed below. 

**IMPORTANT: Please address the following editorial requests: 

1) FINANCIAL DISCLOSURES: Please update your financial disclosures statement in our electronic system to include the grant numbers for all grants, the URL of each funder website, and a statement indicating if the sponsors or funders played any role in the study design, data collection and analysis, decision to publish, or preparation of the manuscript.

2) DATA: I see that you indicate that you will provide your mass spec data, glycan data, and EM images via various repositories and in the supplement. Please do go ahead and do that at this stage, as these data will need to be made available before we can publish your study. Please ensure that data deposited in a repository has a DOI, and provide that to us in your data availability statement. 

3) DATA: In addition to the data listed above, to be compliant with our data policy, we will need you to provide the underlying data for all the other figures in your paper. (See the full PLOS Data Policy, which requires that all data be made available without restriction, here: http://journals.plos.org/plosbiology/s/data-availability) 

Note that for most of the figures, we would not require all raw data. Rather, we ask that all individual quantitative observations that underlie the data summarized in the figures and results of your paper be made available in one of the following forms:

a. Supplementary files (e.g., excel). Please ensure that all data files are uploaded as 'Supporting Information' and are invariably referred to (in the manuscript, figure legends, and the Description field when uploading your files) using the following format verbatim: S1 Data, S2 Data, etc. Multiple panels of a single or even several figures can be included as multiple sheets in one excel file that is saved using exactly the following convention: S1_Data.xlsx (using an underscore).

b. Deposition in a publicly available repository. Please also provide the accession code or a reviewer link so that we may view your data before publication. 

>>Regardless of the method selected, please ensure that you provide the individual numerical values that underlie the summary data displayed in the following figure panels as they are essential for readers to assess your analysis and to reproduce it:

Fig 1C-F; Fig 2A-B; Fig 4A-E; Fig 5A-D;

Fig S1; Fig S4; Fig S5

>>Please also ensure that figure legends in your manuscript include information on where the underlying data can be found, and ensure your supplemental data file/s has a legend.

>>Please ensure that your Data Statement in the submission system accurately describes where your data can be found.

4) CODE: Per journal policy, if you have generated any custom code during the course of this investigation, please make it available without restrictions. Please ensure that the code is sufficiently well documented and reusable, and that your Data Statement in the Editorial Manager submission system accurately describes where your code can be found. 

5) METHODS: I noticed that some of the methods are included in the supplemental material. Please move these to the main text. 

We expect to receive your revised manuscript within two weeks. 

*Published Peer Review History*

*Press*

Sincerely,

Luke

Lucas Smith, Ph.D.

Senior Editor

lsmith@plos.org

PLOS Biology

Reviewer remarks:

Reviewer #1 (who also agreed to serve as the Academic Editor for this study): The authors have sufficiently addressed my comments. Although some points remain to be clarified, I believe the evidence they present should be shared with the community.

Reviewer #2, Ulf Dettmer (note, reviewer 2 has signed this review): The authors did a good job addressing questions from all reviewers.

My key comment #7 has only been partially answered, but future studies will hopefully clarify things further.

Best of luck with your research.

Ulf

---

## [Editor Report · Decision Letter 2]

Dear Janet,

Thank you for the submission of your revised Research Article "Plasma Membrane Remodelling in GM2 Gangliosidoses Drives Synaptic Dysfunction" for publication in PLOS Biology and thank you for addressing our last editorial requests in this revision. On behalf of my colleagues and the Academic Editor, Giovanni D'Angelo, I am pleased to say that we can in principle accept your manuscript for publication, provided you address any remaining formatting and reporting issues. These will be detailed in an email you should receive within 2-3 business days from our colleagues in the journal operations team; no action is required from you until then. Please note that we will not be able to formally accept your manuscript and schedule it for publication until you have completed any requested changes.

PRESS

Sincerely, 

Luke

Lucas Smith, Ph.D.

Senior Editor

PLOS Biology

lsmith@plos.org
